# Recombinant BCG to Enhance Its Immunomodulatory Activities

**DOI:** 10.3390/vaccines10050827

**Published:** 2022-05-23

**Authors:** Magdalena Kowalewicz-Kulbat, Camille Locht

**Affiliations:** 1Department of Immunology and Infectious Biology, Institute of Microbiology, Biotechnology and Immunology, Faculty of Biology and Environmental Protection, University of Lodz, 90-237 Lodz, Poland; camille.locht@pasteur-lille.fr; 2CHU Lille, Institut Pasteur de Lille, U1019–UMR9017–CIIL–Center for Infection and Immunity of Lille, University Lille, CNRS, Inserm, F-59000 Lille, France

**Keywords:** rBCG, cytokines, tuberculosis, bacterial toxins, non-tuberculosis diseases

## Abstract

The bacillus Calmette–Guérin (BCG) is an attenuated *Mycobacterium bovis* derivative that has been widely used as a live vaccine against tuberculosis for a century. In addition to its use as a tuberculosis vaccine, BCG has also been found to have utility in the prevention or treatment of unrelated diseases, including cancer. However, the protective and therapeutic efficacy of BCG against tuberculosis and other diseases is not perfect. For three decades, it has been possible to genetically modify BCG in an attempt to improve its efficacy. Various immune-modulatory molecules have been produced in recombinant BCG strains and tested for protection against tuberculosis or treatment of several cancers or inflammatory diseases. These molecules include cytokines, bacterial toxins or toxin fragments, as well as other protein and non-protein immune-modulatory molecules. The deletion of genes responsible for the immune-suppressive properties of BCG has also been explored for their effect on BCG-induced innate and adaptive immune responses. Most studies limited their investigations to the description of T cell immune responses that were modified by the genetic modifications of BCG. Some studies also reported improved protection by recombinant BCG against tuberculosis or enhanced therapeutic efficacy against various cancer forms or allergies. However, so far, these investigations have been limited to mouse models, and the prophylactic or therapeutic potential of recombinant BCG strains has not yet been illustrated in other species, including humans, with the exception of a genetically modified BCG strain that is now in late-stage clinical development as a vaccine against tuberculosis. In this review, we provide an overview of the different molecular engineering strategies adopted over the last three decades in order to enhance the immune-modulatory potential of BCG.

## 1. Introduction

Bacillus Calmette–Guérin (BCG) vaccination has now been practiced for over 100 years for the prevention of tuberculosis [1]. It is today the oldest vaccine still in use and remains the most widely used vaccine in the world. Although effective against the most severe and deadly forms of the disease, including disseminated tuberculosis, in children [2], its effectiveness against the most prevalent, pulmonary form in adults is still too limited to control the disease, and tuberculosis remains the primary cause of global mortality due to a single infectious agent, *Mycobacterium tuberculosis* [3], except for COVID-19 during the 2020/2021 pandemic. 

As soon as BCG became universally available, observational studies showed that, in addition to providing protection against severe childhood tuberculosis, BCG vaccination also substantially reduced all-cause childhood mortality, beyond what was expected through protection against tuberculosis [4]. These observations have been confirmed by many subsequent studies (for review, see [5]). In addition, more recent studies have shown protective and therapeutic effects of BCG against a variety of heterologous infections [6] and inflammatory and auto-immune diseases [7], including type 1 diabetes [8], multiple sclerosis [9], Alzheimer’s and Parkinson’s diseases [10], as well as various cancers, such as cutaneous melanoma [11] and recurrent non-muscle-invasive bladder cancer (NMIBC) [12]. However, as for protection against tuberculosis, there is room for improvement also for the non-specific effects of BCG.

The advent of molecular engineering technologies adapted for BCG in the early 1990s [13] has provided opportunities to improve BCG efficacy, both for protection against tuberculosis and for its non-specific effects. The latter are likely due to the training of innate immune cells by BCG resulting in increased cytokine production [14]. BCG-induced cytokine responses, such as IFN-γ [15] and IL-17 [16], are also known to be required for protection against tuberculosis. One approach to enhancing the protective performance of BCG is therefore to construct recombinant BCG (rBCG) strains that produce immunomodulatory molecules to enhance its capacity to stimulate innate immune cells. This strategy has been adopted by several laboratories over the last three decades. In this review, we summarize the currently available literature on rBCG strains that produce cytokines, bacterial toxins or other immunomodulatory molecules and on their protective effects against tuberculosis and other diseases, compared to their parental BCG (pBCG) strains. As this review focuses on the enhancement of the immune-modulatory properties of BCG, we did not include rBCG strains producing antigens from heterologous pathogens to engineer specific multivalent vaccines, since they have recently been discussed in the excellent review paper by Mouhoub et al. [17].

## 2. Recombinant BCG Strains Producing Cytokines

The first rBCG strains constructed in order to enhance its immunomodulatory potential were engineered to produce mammalian cytokines, intended to augment the protective potency of BCG against tuberculosis or to increase its potential as an anti-cancer drug, especially against NMIBC. The cytokines were primarily Th1 cytokines, including IL-2, IFN-α, IFN-γ, IL-7, IL-12, IL-15, IL-18, IL-21, IL-35 and TNF-α (Table 1). 

### 2.1. IL-2

IL-2 was the first cytokine produced by rBCG. This cytokine was originally identified as a T-cell growth factor, but later its pleiotropic effects on different immune cells were uncovered [18]. It is pivotal in the induction of immune responses by enhancing the expansion and activation of CD4^+^, CD8^+^ and NK cells [18]. Furthermore, IL-2 has been proposed as a treatment for mycobacterial infections, and the administration of IL-2 to mice was shown to increase their resistance to mycobacterial infections. [19]. 

Biologically active IL-2 could be produced in rBCG (rBCG::IL-2) by placing the genes encoding murine and rat IL-2 under the control of the BCG *hsp60* promoter. When expressed as a fusion with the alpha antigen signal peptide, IL-2 was secreted into the extracellular milieu [20]. Mouse splenocytes stimulated in vitro with rBCG::IL-2 produced significantly more IFN-γ than splenocytes stimulated with pBCG. Human IL-2 was also produced by rBCG, using a similar approach (rBCG::hIL-2) [21]. Upon intravenous injection into BALB/c mice, the number of rBCG::hIL-2 bacteria present in the spleen was significantly reduced compared to the control mice injected with pBCG, which the authors attributed to a probable augmentation of the host immune response by the secreted IL-2 [21]. Similar findings were later made by using rBCG strains producing green fluorescent protein in addition to murine IL-2 [22]. This also resulted in decreased spleen weight of the mice infected with the rBCG strain.

Enhanced mycobacteria-specific cytokine responses could also be demonstrated after intravenous administration of rBCG::IL-2 compared to pBCG. Splenocytes stimulated with purified protein derivative (PPD) produced significantly more IL-2, IFN-γ, IL-10, IL-3 and GM-CSF 16 weeks after vaccination with rBCG::IL-2 than with pBCG [23]. This difference was also seen when PPD-specific splenocyte proliferation was measured. In this study, no IL-4 release was detected in any group. However, another study [24] found that IL-4 expression was lower in red deer vaccinated with the same rBCG::IL-2 compared to pBCG, as measured by mRNA levels. The rBCG::IL-2 strain also induced lower delayed-type hypersensitivity upon intradermal injection of mycobacterial antigens. 

Overall, these observations document the enhanced immune-modulatory effects of rBCG::IL-2 over pBCG and a further skewing of the immune response towards a Th 1 type. This was also reflected in the antibody isotype profiling, where rBCG::IL-2 induced a stronger serum IgG2a over IgG1 ratio than pBCG [25]. Importantly, vaccination with rBCG::IL-2 induced longer-lasting Th1 T cell responses in the spleen than vaccination with pBCG [25]. Whether this translates into improved protection against tuberculosis or other diseases was not examined in these studies. However, Young et al. [25] have shown that nasal infection with BCG 16 weeks after subcutaneous vacation resulted in a lower bacterial burden in the spleen in the rBCG::IL-2-vaccinated compared to the pBCG-vaccinated mice. In contrast, when virulent *Mycobacterium bovis* was used in the challenge, no significant difference in protective efficacy between the two vaccine strains could be detected. 

However, superior protection afforded by rBCG::IL-2 over pBCG may still be possible in a model in which pBCG provides poor protection, such as in the context of exacerbated Th2 responses or Th1 type suppression. Splenocytes from dexamethasone (DXM)-treated mice proliferated more strongly and secreted higher levels of IFN-γ after vaccination with rBCG::IL-2 than with pBCG [26]. rBCG::IL-2-vaccinated mice also induced more CD8^+^ and activated CD4^+^ T cells than pBCG-vaccinated mice, indicating that rBCG::IL-2 was able to evoke a strong Th1 type immune response in an immunocompromised host, whereas pBCG did not. Similarly, in transgenic mice overproducing IL-4, rBCG::IL-2 could change the IL-4-dependent Th2 profile into a Type 1 immune response, while pBCG could not. This was accompanied by an increase in the numbers of activated splenic CD4^+^ and CD8^+^ T cells. [26]. Unfortunately, the study did not address a potentially improved protection by rBCG::IL-2 against *M. tuberculosis* challenge in these models.

In an effort to further enhance the protective immunity of rBCG strains against tuberculosis, IL-2 has been fused to Early Secreted Antigenic Target 6 (ESAT-6), a *M. tuberculosis* antigen absent in BCG [27]. ESAT-6 is a protective antigen and its production as a hybrid protein with IL-2 in an rBCG strain was hoped to improve BCG-mediated protection. ESAT-6 was genetically fused to the C-terminal end of human IL-2 and the hybrid protein was produced by rBCG::IL-2-ESAT-6 under the control of the *hsp60* promoter and secreted via the signal peptide of the alpha antigen. As for rBCG::IL-2 above, rBCG::IL-2-ESAT-6 persisted less well in the spleen of mice than pBCG. As expected, only rBCG::IL-2-ESAT-6 induced IFN-γ and cytotoxic T cell responses to ESAT-6 in mice after vaccination. However, it also induced stronger IFN-γ responses to PPD. The protective efficacy of rBCG::IL-2-ESAT-6 against tuberculosis was not investigated in this study. Furthermore, rBCG::IL-2-ESAT-6 was not compared to rBCG::IL-2 or an rBCG strain producing only ESAT-6. Therefore, the potential relative benefits of IL-2 and ESAT-6 remain unknown.

Because of its ability to promote the growth of T cells, enhance NK cell activity and induce IFN-γ and TNF-α secretion, IL-2 may also be useful for anticancer therapy. Consequently, rBCG::IL-2 secreting murine IL-2 [20] has been tested in a mouse model of B16 melanoma [28]. In this model, B16 melanoma tumor cells were implanted into mice, followed by intra-tumoral or subcutaneous injections of rBCG::IL-2 or pBCG. Intra-tumoral injection of either BCG strain significantly reduced tumor size and improved survival of the mice, whereas subcutaneous administrations of the BCG stains did not. However, no difference between the two strains was noted. Similarly, when the BCG strains were administered 14 days before a B16 challenge, both strains resulted in around a 45% reduction in tumor burden [28]. 

### 2.2. IFN-γ

Instead of constructing IL-2-producing rBCG strains, an alternative approach to enhancing Th1 skewing by BCG is to directly express IFN-γ by rBCG in an attempt to improve protection against tuberculosis. It has been well documented that the absence of the IFN-γ or IFN-γ receptor leads to the loss of control of mycobacterial infections, both in mice [29] and in humans [30]. Murine IFN-γ was thus produced by rBCG strains under the control of the *hsp60* promoter and secreted via the alpha antigen signal peptide [23]. Similar to rBCG::IL-2, administration of rBCG::IFN-γ to mice also induced stronger lymphocyte proliferation and cytokine responses than administration of pBCG, despite reduced bacterial numbers in the lungs and spleen [31]. In IFN-γ^−/−^ mice, rBCG::IFN-γ was also cleared more rapidly from the mouse’s lungs while inducing stronger expression of iNOS, an IFN-γ-induced protein, and decreasing IL-10 expression compared to pBCG. However, neither pBCG nor rBCG::IFN-γ was able to protect IFN-γ^−/−^ mice from death after aerosol challenge with *M. tuberculosis.* These observations suggested that the local production of IFN-γ is insufficient for anti-mycobacterial immunity.

Although infection with BCG often leads to the protection of the host, it sometimes causes tissue pathology and active fibrosis due to local dermal immune responses. It is therefore conceivable that rBCG::IFN-γ vaccination may result in exacerbated dermal fibrosis. In a murine model of fibrosis, intravenous infection with BCG caused increases in liver weight and number of granulomas in a dose-dependent fashion. However, liver weight, number of granulomas in the liver and fibrosis measured by the content of hydroxyproline in the liver were reduced when the mice were infected with rBCG::IFN-γ instead of pBCG [32]. These findings suggest that local secretion of IFN-γ by rBCG::IFN-γ causes early activation of protective type I responses, manifested by promoting bacterial killing while reducing the risk of pathological processes and actually preventing fibrosis. 

Like for IL-2 above [27], mouse IFN-γ was also combined with overexpression of mycobacterial antigens, including ESAT-6 and Antigen 85B (Ag85B) in an rBCG strain secreting an Ag86B-ESAT-6-IFN-γ hybrid protein [33]. In a study comparing rBCG::Ag85B-ESAT-6-IFN-γ with rBCG::Ag85B-ESAT-6, rBCG::Ag85B and pBCG for their immunogenicity and protective potency against *M. tuberculosis* challenge, all three rBCG strains performed better than pBCG when the bacterial burden was measured in the lungs 6 and 9 weeks after challenge. There was no difference between the three rBCG strains. However, when the bacterial burden in the spleen was measured, rBCG::Ag85B-ESAT-6-IFN-γ performed better than the other BCG strains. Antibody and T cell responses to PPD were induced at similar levels by all BCG strains, while, as expected, anti-Ag85B and anti-ESAT-6 responses were only induced after vaccination with rBCG strains producing these antigens. 

This study was followed up by a study in which an rBCG strain producing an Ag85B-IFN-γ hybrid protein was evaluated in mice. Vaccination with rBCG::Ag85B-IFN-γ induced more AG85B-specific IFN-γ and TNF-α production by mouse splenocytes and enhanced Ag85B-specific cell proliferation and higher titers of anti-Ag85B antibodies than vaccination with pBCG [34]. The number of Ag85B-specific CD4^+^ T cells was also higher in the rBCG::Ag85B-IFN-γ-vaccinated compared to the pBCG-vaccinated mice, while CD8^+^ T cell numbers were similar in the two groups. rBCG::Ag85B-IFN-γ also induced stronger expression of the co-stimulatory molecules CD80, CD86 and CD40 than pBCG. However, the protective efficacy of rBCG::Ag85B-IFN-γ was not assessed, nor was the added value of IFN-γ over rAg85B alone.

The effect of IFN-γ production by rBCG has also been examined in a bladder cancer model. rBCG::IFN-γ was found to significantly up-regulate the expression of MHC class I molecules on the surface of a murine bladder cancer cell line compared to pBCG [35]. Intravesical instillation of rBCG::IFN-γ also enhanced the recruitment of CD4^+^ T cells into the bladder and increased IL-2 and IL-4 mRNA expression, which resulted in prolonged survival compared to mice treated with pBCG. 

### 2.3. IFN-α

In superficial bladder cancer immunotherapy, IFN-α has been successfully used to treat patients that did not respond to BCG therapy [36]. By adding IFN-α to BCG, the dose of BCG could be lowered to efficiently treat bladder cancer in a murine model, so that BCG’s toxicity was strongly reduced, while the activity against tumors was increased [37]. Consequently, IFN-α has also been produced by rBCG, in particular with the aim to overcome the relatively short half-life of IFN-α instilled into the bladder. Luo et al. [38] constructed an rBCG strain that constitutively secretes human IFN-α (rBCG::hIFN-α) under the control of the *hsp60* promoter and the alpha antigen signal peptide, and found that compared to pBCG, rBCG::hIFN-α strongly enhanced IFN-γ production by human peripheral blood mononuclear cells (PBMC), while it decreased IL-10 production. rBCG::hIFN-α also accelerated IFN-γ production and the study revealed a strong synergy between BCG and IFN-α for the induction of IFN-γ. 

The anti-tumor effect of rBCG::hIFN-α was demonstrated by the release of lactate dehydrogenase from tumor cells incubated with pBCG- or rBCG::hIFN-α-activated killer cells, which showed a stronger anti-tumor effect of rBCG::hIFN-α than of pBCG [39,40,41]. These killer cells can be generated by stimulating human PBMC with viable BCG strains, and they can kill bladder cancer cells through perforin-mediated lysis. The generation of these killer cells by rBCG::hIFN-α relied on IFN-α-mediated induction of IFN-γ and IL-2, as cytotoxicity was decreased in the presence of neutralizing antibodies to IFN-γ, IL-2 or IFN-α [42]. The cytotoxic action of rBCG::IFN-α-stimulated PBMC was mainly due to NK cells, and to a minor extent to CD8^+^ T cells.

The direct effect of rBCG::hIFN-α on bladder cancer cells has been demonstrated by morphological changes in the MB49 cancer cell line, slower growth of these cells and induction of apoptosis upon co-culturing with rBCG::hIFN-α [43]. In a murine model of orthotopic bladder cancer, rBCG::hIFN-α induced bladder inflammation with strong infiltration of CD3^+^ lymphocytes, CD20^+^ monocytes and Gr1^+^ polymorphonuclear cells, and rBCG::hIFN-α-treated mice survived significantly longer than pBCG-treated mice.

### 2.4. IL-18

One of the cytokines which plays a key role in the expression of cell-mediated immunity against mycobacteria is IL-18. This cytokine has been originally named IFN-γ-inducing factor and acts in synergy with IL-12 to induce IFN-γ production by B, T and NK cells [44]. When secreted under the control of the *hsp60* promoter and the alpha-antigen signal peptide by rBCG murine, IL-18 strongly enhanced BCG-induced PPD-specific IFN-γ responses after vaccination of BALB/c and C3H/HeJ mice [45]. GM-CSF and IL-10 production were also augmented, while the IgG levels in serum were lower in rBCG::mIL-18-vaccinated mice compared to pBCG, indicating further Th1 skewing by the secretion of mIL-18. However, BCG-induced protection of mice against *M. tuberculosis* challenge was not increased by the secretion of mIL-18. In fact, rBCG::mIL-18 appeared to be somewhat less protective than pBCG [25]. Similarly, in a different BCG genetic background, the expression of IL-18 by rBCG did not improve protection over the parental strain [46]. Secretion of human IL-18 in an rBCG strain that already expressed listeriolysin (VPM1002, see below) did not further decrease bacterial burden in the lungs and spleen of mice upon *M. tuberculosis* infection compared to VPM1002 vaccination alone, although it significantly augmented the production of pro-inflammatory cytokines, such as TNF-α, IL-12p70, GM-CSF, IL-6, IP-10, RANTES and Eotaxin, and enhanced antigen-specific CD4^+^ T cell responses.

These findings were confirmed by an independent study [47], which further showed that IFN-γ production could be abolished by neutralizing anti-IL-18 or anti-IL-12 antibodies, confirming the synergistic nature of IFN-γ induction by IL-12 and IL-18, while anti-IL-10 antibodies augmented IFN-γ production. Although Biet et al. [45] did not detect reduced growth in the spleen of BALB/c and C3H/HeJ mice after rBCG::mIL-18 infection compared to pBCG, reduced growth was seen in the spleen of C57BL/6 mice [47]. Furthermore, in contrast to BALB/c mice [45], splenocytes from C57BL/6 mice produced less IL-10 after rBCG::mIL-18 compared to pBCG vaccination [47], indicating mouse strain dependence of the IL-18 effect on IL-10 production. Although the study by Luo et al. [47] did not examine the protective effect of rBCG::mIL-18 against *M. tuberculosis*, it did assess its effect against bladder cancer tumor cells. While peritoneal exudate cells stimulated with pBCG showed weak cytotoxicity against a bladder cancer cell line, rBCG::mIL-18-stimulated cells expressed strong killing of the cancer cells, and this could be inhibited by the addition of anti-IL-12 or anti-IL-18 antibodies.

Although the strong capacity of IL-18 in synergy with IL-12 to enhance Th1 type responses would be expected to down-regulated Th2 responses, high doses of IL-18 were reported to increase allergic sensitization, Th2 cytokine production and airway eosinophilia in a ragweed mouse model of allergic asthma, which was reversed in IFN-γ^−/−^ mice [48]. In contrast, in an ovalbumin mouse model of allergic asthma, a reduction in the ovalbumin-specific Th2 cytokine IL-5 production by lung draining lymph node cells was observed in rBCG::mIL-18-vaccinated mice compared to non-vaccinated or pBCG-vaccinated mice, and this was paralleled by an increase in IFN-γ production by these cells [49]. Broncho-alveolar eosinophilia was also inhibited more strongly by rBCG::mIL-18 than by pBCG in this model, while macrophage numbers were not affected by either BCG strain. These conflicting results may be due to the lower IL-18 amounts administered via rBCG::mIL-18 over several weeks compared to the high peak amounts that were given in the study by Wild et al. [48], as it is conceivable that the action of IL-18 may differ according to the dose.

In a human in vitro model, stimulation of human monocyte-derived dendritic cells (DCs) from healthy donors who had been BCG immunized in their childhood with an rBCG strain producing human IL-18 (rBCG::hIL-18) resulted in increased IL-23 and IL-10 production, compared to pBCG-stimulated DCs, while the expression of the surface markers CD80 and CD86 was not modified [50]. rBCG::hIL-18-stimulated human DCs also triggered strong IFN-γ secretion by co-cultured naive CD4^+^ T cells, but not memory CD4^+^ T cells, which was accompanied by moderately elevated IL-10 levels. The enhanced IFN-γ induction by rBCG::mIL-18 was abolished in the presence of anti-IL-18 antibodies.

IL-10 and IP-10 were also induced by rBCG::hIL-18 in DCs from allergic patients sensitive to Der p 1, which is the main allergen of the house dust mite, whereas IL-23 was not induced in the DCs from these patients [51]. Importantly, stimulation of DCs with rBCGhIL-18 in the presence of Der p 1 decreased IL-5 secretion by naive T cells to a larger extend than pBCG-stimulated DCs. Thus, rBCG strains producing IL-18 express enhanced immunomodulatory effects both in mice and in humans.

### 2.5. IL-15

IL-15 is a pro-inflammatory cytokine with structural similarities to IL-2 [52]. It is produced by monocytes, macrophages, DCs and skeletal muscle cells in the kidneys, lungs and the heart, and has multi-directional effects on the immune system [53]. IL-15 is responsible for the generation of NK cells; it increases their cytotoxic activity and regulates the secretion of IFN-γ [54]. It also activates neutrophils and macrophages, leads to the proliferation of T lymphocytes and supports their activation, and plays an important role in the maintenance of memory CD8^+^ T cells [55]. It might therefore be an important component for the induction of efficient cell-mediated immunity to intracellular pathogens, including *M. tuberculosis*. To examine whether the protective effect of BCG against *M. tuberculosis* can be increased by the secretion of IL-15, an rBCG secreting murine IL-15 (rBCG::IL-15) was constructed [56]. Compared to pBCG, it was cleared faster from the spleen of mice but induced stronger NK-mediated cytotoxicity and IFN-γ responses by splenocytes and higher CD8^+^ memory T cell numbers. Curiously, while pBCG administration to mice primed with a *Listeria monocytogenes* strain that produces ovalbumin peptides eroded the CD8^+^ T cell responses to these peptides, administration of rBCG::IL-15 completely abolished this CD8^+^ T cell response, regardless of the timing between priming and rBCG::IL-15 infection, indicating that rBCG::IL-15 induces attrition rather than maintenance of pre-existing CD8^+^ T cells. This was also reflected by a decrease in anti-tumor activity of the recombinant *L. monocytogenes* when rBCG::IL-15 was administered. Thus, while on the one hand rBCG::IL-15 induced potent activation of CD8^+^ T cells and NK cells, it also facilitated the erosion of pre-existing CD8^+^ T cell memory.

Nevertheless, in another study, rBCG::IL-15 administration was shown to significantly prolong the survival of mice inoculated with bladder tumor cells and suppress tumor growth compared to mice treated with pBCG [57]. This effect was accompanied by increased infiltration of neutrophiles and expression of the chemokines MIP-2 and MIP-1α, and depletion of neutrophils resulted in the loss of rBCG::IL-15-enhanced survival.

The effect of IL-15 on protection against tuberculosis by rBCG was also investigated. When Ag85B was fused to IL-15 and secreted by an rBCG strain (rBCG::Ag85B-IL-15), protection against challenge with *M. tuberculosis* in mice was improved over vaccination with an rBCG strain over-producing only Ag85B (rBCG::Ag85B), as demonstrated by a lower bacterial burden in the lungs and less lung pathology [58]. Immunization with rBCG::Ag85B-IL-15 also led to more CD44^+^CD8^+^ T and CD44^+^CD4^+^ T cells expansion and higher frequencies of antigen-specific IFN-γ-producing T cells compared to rBCG::Ag85B, despite faster clearance.

### 2.6. IL-12

As IL-12 can augment both innate and cellular immunity and can strengthen Th1-type immune responses against *M. tuberculosis*, especially in synergy with IL-18, this cytokine was also a target for the development of rBCG. Mice vaccinated with an rBCG strain that produces IL-12 together with CFP-10 and Ag85B generated a more efficient immune response compared to pBCG or rBCG::Ag85B–CFP10 [59]. Especially polyfunctional CD4^+^ and CD8^+^ T cells producing IL-2, IFN-γ and/or TNF-α were induced at higher levels in the spleen and lungs after rBCG::Ag85B–CFP10-IL-12 vaccination than after vaccination with rBCG::Ag85B–CFP10 or pBCG. Furthermore, the splenocytes from rBCG::Ag85B–CFP10-IL-12-vaccinated mice expressed higher cytotoxicity against macrophages infected with *M. tuberculosis* than splenocytes from mice vaccinated with rBCG::Ag85B–CFP10 or with pBCG. In a follow-up study, superior IFN-γ induction by rBCG::Ag85B–CFP10-IL-12 over rBCG::Ag85B–CFP10 and pBCG was confirmed, as well as its superior ability to induce cytotoxic T cells against *M. tuberculosis*-infected macrophages [60]. This study extended the previous observations by showing that peripheral blood mononuclear cells from rBCG::Ag85B–CFP10-IL-12-vaccinated mice also produced more TNF-α upon stimulation with *M. tuberculosis*, and even upon stimulation with an unrelated pathogen, such as *Pseudomonas aeruginosa*, than cells from pBCG- or rBCG::Ag85B–CFP10-vaccinated mice.

In a similar study, IL-12 was co-expressed with ESAT-6 in rBCG [61]. This strain, rBCG::ESAT-6-IL-12, also induced stronger T cell responses to mycobacterial antigens than pBCG or rBCG::ESAT-6, as documented by T cell proliferation assays and IFN-γ measurements upon stimulation of spleen cells with PPD. Surprisingly, IL-12 expression alone in rBCG::IL-12 did not enhance T cell responses over pBCG. Furthermore, IL-12 expression, with or without ESAT-6, appeared to diminish the protective potential of BCG against *M. tuberculosis* infection, as evidenced both by bacterial counts in the lungs and spleen and by pathology scoring in the lungs.

In another study, in which IL-12 was co-expressed with Ag85A in rBCG::IL-12-Ag85A, the recombinant strain also induced stronger T cell proliferation and IFN-γ responses than pBCG, rBCG::Ag85A or rBCG::IL-12 [62]. In this case, rBCG::IL-12-Ag85A induced similar levels of protection as pBCG against *M. tuberculosis* challenge, but curiously, rBCG::IL-12 and rBCG::Ag85A provided less protection than pBCG with respect to bacterial counts in the lungs and spleen.

A more recent study combined the three antigens Ag85A, CFP-10 and ESAT-6 together with IL-12 and GM-CSF in various combinations or individually [63]. Immunogenicity studies indicated that IgG, IgG1 and IgG2a titers and IFN-γ levels secreted by splenocytes were strongest when IL-12 or GM-CSF was co-expressed with the mycobacterial antigens. When protection against *M. tuberculosis* was evaluated, all recombinant strains provided significant protection when compared to non-vaccinated controls, but the co-expression of IL-12 or GM-CSF did not significantly augment the protective effect of the rBCG strains.

### 2.7. GM-CSF

GM-CSF is one of the strongest regulatory cytokines of DCs and their antigen-presenting function [64]. Murine GM-CSF could be secreted by rBCG::GM-CSF using the same *hsp60* promoter/alpha antigen signal peptide system as used for other cytokines, and the recombinant strain was shown to induce various cytokines at levels similar or superior to those induced by rBCG::IL-2 after intravenous or intraperitoneal infection of mice [23]. Subcutaneous immunization with rBCG::GM-CSF led to increased antigen-presenting cell numbers in the draining lymph nodes of mice, compared to immunization with pBCG [65]. rBCG::GM-CSF vaccination of mice also enhanced the expression of co-stimulatory molecules on DCs in the draining lymph nodes, which was associated with increased and sustained numbers of PPD-specific IFN-γ-secreting T cells. When the mice were aerosol-challenged with *M. tuberculosis*, rBCG::GM-CSF-vaccinated mice had consistently lower bacterial loads in the lungs and spleen than pBCG-vaccinated mice.

Improved immunogenicity of a GM-CSF-producing rBCG strain was also seen with an rBCG producing a GM-CSF-ESAT-6 hybrid protein [66]. Antibody responses to PPD, as well as PPD-specific T cell responses, including the induction of specific CD4^+^ and of CD8^+^ T cells and IFN-γ responses, were strongest in mice immunized with rBCG::GM-CSF-ESAT-6 compared to mice immunized with rBCG::GM-CSF, rBCG::ESAT-6 or pBCG. Protection against *M. tuberculosis* infection was not evaluated in that study.

rBCG::GM-CSF strains have also been explored for cancer therapy. Mucin 1 (MUC1) is a tumor-associated antigen expressed on a variety of epithelial adenocarcinomas, including breast cancers. An rBCG strain that produces a hybrid protein composed of repeated MUC1 epitopes fused to GM-CSF (rBCG::MUC1-GM-CSF) was found to significantly limit tumor growth and prolong the survival of mice inoculated with MUC1^+^ breast cancer cells, while a pBCG or rBCG::MUC1-GM-CSF strain containing a single repeat of the epitope did not [67], indicating the predominant role of the MUC1 epitope over GM-CSF in tumor growth inhibition and mouse survival. Improved survival and tumor growth inhibition was paralleled with enhanced IFN-γ production and CD8^+^ T cell responses.

Specific targeting of GM-CSF produced by rBCG was attempted by engineering a strain that produces human GM-CSF fused to BZLF1 in an Epstein–Barr virus (EBV) [68]. In concomitant EBV infection and cancer, only the cancer cells are infected with EBV and therefore express BZLF1. Treatment of EBV^+^ tumor cells-inoculated mice with rBCG::hGM-CSF-BZLF1 significantly delayed tumor formation and prolonged mouse survival compared to treatment with pBCG, rBCG::hGM-CSF or rBCG::BLZF1, most likely through the induction of cytotoxic T lymphocytes.

Another method of targeting EBV-associated carcinoma cells was explored through the construction of an rBCG strain that produces hGM-CSF fused to LMP2A, an EBV latent membrane protein that is recognized by cytotoxic T cells [69]. Immunization of mice with rBCG::hGM-CSF-LMP2A induced cytotoxic T cell responses able to lyse EBV^+^ tumor cells with significantly higher efficacy than immunization with pBCG. As a consequence, tumor growth was slower in rBCG::hGM-CSF-LMP2A-treated mice than in pBCG-, rBCG::hGM-CSF- or rBCG::LMP2A-treated mice. Tumor size and weight were also significantly decreased in the former compared to the latter groups. This was concomitant with increased CD4^+^ T cell proliferation in the tumor.

### 2.8. Other Cytokines

Other cytokines which were used as components of rBCG constructs include TNF-α, IL-7, IL-21 and IL-35.

As TNF-α plays an important role in host defense against *M. tuberculosis* by contributing to the formation of granulomas [70], it was an obvious cytokine to be produced by rBCG. An rBGC strain expressing Ag85B-ESAT-6-TNF-α was found to induce stronger immune responses than rBCG::Ag85B-ESAT-6 or pBCG [71], as shown by augmented PPD-specific and Ag85B-specific IFN-γ responses 8 weeks after vaccination. Antibody responses to Ag85B were also increased in rBCG::Ag85B-ESAT-6-TNF-α-vaccinated compared to rBCG::Ag85B-ESAT-6-vaccinated mice, as was the IgG2c to IgG1 ratio, indicative of Th1 skewing. Protection by rBCG::Ag85B-ESAT-6-TNF-α against *M. tuberculosis* challenge was not investigated in this study.

IL-7 plays a key role in the development and maintenance of IL-17A^+^γδ T cells, which are present in mucosal tissues, including the lungs, and produce IL-17 at the early stage of mycobacterial infections in murine models [72,73]. IL-7 is important for the formation of granulomas and for the induction of Th1 response against mycobacterial infections [72]. An rBGC strain expressing a chimeric protein composed of Ag85B fused to IL-7 (rBCG::Ag85B-IL-7) was found to increase the numbers of IL-17A^+^γδ T cells and mycobacteria-specific Th1 cells in the lungs compared to rBCG::Ag85B [74]. No effect was seen on IFN-γ^+^γδ T cell numbers. However, PPD-specific and Ag85B-specific IFN-γ-producing CD4^+^ T cell numbers in the lungs were higher in rBCG::Ag85B-IL-7-vaccinated mice than in rBCG::Ag85B-vaccinated mice. These responses were abolished in *vg4/6^−/−^* mice, confirming the essential role of the γδ T cells in the rBCG::Ag85B-IL-7-induced Th1 response. However, protection against *M. tuberculosis* was not enhanced by the expression of IL-7 in rBCG [46].

To examine the role of IL-21 in the regulation of IL-17A^+^γδ T cells by BCG, an rBCG::AG85B-IL-21 strain was compared to rBCG::Ag85B [75]. The numbers of IL-17A^+^Vγ6^+^ γδ T cells were significantly lower after inoculation with rBCG::Ag85B-IL-21 compared to rBCG::Ag85B. This was consistent with the observation that IL-17A^+^Vγ6^+^ γδ T cell numbers were increased in the peritoneal cavity of IL-21R KO mice compared to wild-type mice after BCG infection. Apparently, exogenous IL-21 selectively induced apoptosis of IL-17A^+^Vγ6^+^ γδ T cells by a mechanism that still needs to be elucidated. Thus, IL-21 appears to act as an inhibitor of IL-17A^+^ γδ T-cells during BCG infection.

Very recently, IL-35 was also produced in rBCG [76]. This cytokine belongs to the IL-12 family and is capable of suppressing T cell proliferation and inducing IL-35-producing induced regulatory T (Treg) cells to limit inflammatory responses [77]. Therefore, an rBCG::IL-35 strain was constructed as a potential anti-asthma agent. In an RSV-induced asthmatic newborn mouse model, rBCG::IL-35 significantly limited lung tissue injury and the deposition of collagen fibers [76]. It also reduced serum IgE levels and the numbers of inflammatory cells in bronchoalveolar lavage fluids of asthmatic mice. While Th17 cell differentiation was inhibited by rBCG::IL-35, Treg cell numbers were increased, potentially explaining the reduction of allergic airway inflammation in rBCG::IL-35-treated mice.

**Table 1 vaccines-10-00827-t001:** Recombinant BCG producing cytokines in tuberculosis and non-tuberculosis diseases.

Name of rBCG Strain	Experimental Model	Main Findings	Reference
**Tuberculosis**			
**rBCG::hIL-2**	BALB/c	increased production of IFN-γ by splenocytes reduced numbers of rBCG::hIL-2 in the spleen,	[21]
**rBCG::IL-2/GFP** ^a^	C57BL/6	diminished bacterial counts and decreased spleen weight augmented protective immunity to mycobacterial infection via an enhanced Th1 response accelerated macrophage maturation by IFN-γ-production by Th1 or/and NK cells	[22]
**rBCG::IL-2**	BALB/c, C57BL/6, C3H/HeJ	increased IL-2, IFN-γ, IL-10, IL-3, GM-CSF production; lack of IL-4 by PPD-stimulated splenocytes 16 weeks after vaccination	[23]
**rBCG::IL-2**	red deer	decreased IL-4 production lower DTH upon intradermal administration of mycobacterial antigens	[24]
**rBCG::IL-2**	BALB/c, C57BL/6, C3Hej, IL-4 Tg	stronger serum IgG2a over IgG1 ratio high Ag-specific IFN-γ:IL-4 ratio induced longer-lasting Th1 T cell responses in the spleen lower bacterial burden in the spleen	[25]
**rBCG::IL-2**	IL-4 Tg	enhanced proliferative response of splenocytes higher levels of IFN-γ low levels of IgG1 more CD8^+^ and activated CD4^+^ T cells lack of improved protection against *M. tuberculosis* challenge	[26]
**rBCG::IL-2-ESAT-6**	BALB/c	less persistence in the spleen induction of IFN-γ and cytotoxic T cell responses stronger IFN-γ responses to PPD	[27]
**rBCG::IFN-γ**	BALB/c, IFN-γ^−/−^	stronger lymphocyte proliferation and cytokine responses reduced bacterial numbers in lungs and spleen stronger expression of iNOS decreased IL-10 expression lack of protection from death after aerosol challenge with *M. tuberculosis*	[31]
**rBCG::Ag85B-ESAT-6-IFN-γ**	C57BL/6	improved immunogenicity and protective potency against *M. tuberculosis* no differences in antibody and T cell responses to PPD	[33]
**rBCG::Ag85B-IFN-γ**	C57BL/6	enhanced production of AG85B-specific IFN-γ and TNF-α by mouse splenocytes enhanced Ag85B-specific cell proliferation higher titers of anti-Ag85B antibodies increased numbers of Ag85B-specific CD4^+^ T cells stronger expression of the co-stimulatory molecules CD80, CD86 and CD40	[34]
**rBCG::hIFN-α**	human buffy coat	strongly enhanced IFN-γ production by PBMC; decreased IL-10 production; accelerated IFN-γ production	[38]
**rBCG::IL-18** **rBCG::mIL-18**	BALB/c and C3H/HeJ, C57BL/6	enhanced BCG-induced PPD-specific IFN-γ responses after vaccination augmented GM-CSF and IL-10 production lower IgG levels in serum lack of improved protection against *M. tuberculosis* less IL-10 produced by splenocytes	[45,47]
**rBCG::hIL-18**	human DCs	increased IL-23 and IL-10 production by DCs strong IFN-γ secretion by co-cultured naive CD4^+^ T cells, but not memory CD4^+^ T cells moderately elevated IL-10	[50]
**rBCG::IL-15**	C57BL/6	cleared faster from the spleen induced strong NK-mediated cytotoxicity, IFN-γ responses by splenocytes and higher CD8^+^ memory T cell numbers facilitated erosion of pre-existing CD8^+^ T cell memory	[56]
**rBCG::IL-15**	human PBML	activates neutrophils and macrophages, leads to the proliferation of T lymphocytes, supports their activation, and plays an important role in the maintenance of memory CD8^+^ T cells	[55]
**rBCG::Ag85B-IL-15**	C57BL/6	improved protection against tuberculosis (evidenced by a lower bacterial burden in the lungs and less lung pathology more CD44^+^CD8^+^ T and CD44^+^CD4^+^ T cells expansion and higher frequencies of antigen-specific IFN-γ-producing T cells	[58]
**rBCG::Ag85B–CFP10-IL-12**	C57BL/6 C3H/HeJ	PBML produced more TNF-α upon stimulation with *M. tuberculosis* Significantly improved protection against TB	[60]
**rBCG::IL-12-Ag85A**	BALB/c	stronger T cell proliferation and IFN-γ responses similar levels of protection as pBCG against *M. tuberculosis* challenge	[62]
**rBCG producing various combinations of Ag85A, ESAT6 and CFP10, with or without GM-CSF or IL-12**	BALB/c	IgG, IgG1 and IgG2a titers, and IFN-γ levels secreted by splenocytes were strongest when IL-12 or GM-CSF were co-expressed with the mycobacterial antigens significant protection against tuberculosis co-expression of IL-12 or GM-CSF did not significantly augment the protective effect	[63]
**rBCG::GM-CSF**	C57BL/6 B6.SJL/PtprC	increased antigen-presenting cell numbers in the draining lymph nodes enhanced expression of co-stimulatory molecules on DCs in the draining lymph nodes lower bacterial loads in the lungs and spleen after *M. tuberculosis* challenge	[65]
**rBCG::GM-CSF-ESAT-6**	BALB/c	stronger antibody responses increased induction of specific CD4^+^ and of CD8^+^ T cells and IFN-γ responses	[66]
**rBCG::Ag85B-ESAT-6-TNF-α**	C57BL/6	augmented PPD-specific and Ag85B-specific IFN-γ responses increased antibody responses to Ag85B Th1 skewing	[71]
**rBCG::Ag85B-IL-7**	C57BL/6	increased numbers of IL-17A^+^γδ T cells in the lungs no effect on IFN-γ^+^γδ T cell numbers protection against *M. tuberculosis* was not enhanced	[74]
**rBCG::AG85B-IL-21**	C57BL/6	lower numbers of IL-17A^+^Vγ6^+^ γδ T cells compared to rBCG::Ag85B IL-21 appears to act as an inhibitor of IL-17A^+^ γδ T-cells during BCG infection	[75]
**Non-Tuberculosis Diseases**			
**rBCG::IL-2**	C57BL/6	**B16 melanoma tumor cell model**—significantly reduced tumor size and improved survival of the mice	[28]
**rBCG::IFN-γ**	BALB/c, IFN-γ^−/−^	**model of fibrosis**—reduced liver weight, numbers of granulomas in the liver and fibrosis measured by the content of hydroxyproline	[32]
	C57BL/6, Tumor cell line MB49	**bladder cancer**—up-regulated expression of MHC class I molecules on the surface of a murine bladder cancer cell line; enhanced recruitment of CD4^+^ T cells into the bladder and increased IL-2 and IL-4 mRNA expression	[35]
**rBCG::IFN-α**	Tumor cell line MB49	**bladder cancer**—strongly reduced BCG toxicity; increased activity against tumors	[37]
	Tumor cells	**bladder cancer**—stronger anti-tumor effect, decreased cytotoxicity	[39,40,41]
**rBCG::hIFN-α**	Tumor cell line MB49	slower growth of the cells and induction of apoptosis	[43]
	MB49 Tumor cell line MB49	**bladder cancer**—induced bladder inflammation with strong infiltration of CD3^+^ lymphocytes, longer survival of rBCG::hIFN-α-treated mice compared to parental BCG-treated animals	
**rBCG::mIL-18**	Bladder cancer cell line MBT-2	**bladder cancer**—strong killing of the cancer cells	[47]
**rBCG::mIL-18**	BALB/c	**OVA-dependent allergic asthma**—decrease of IL-5; increased IFN-γ production inhibited broncho-alveolar eosinophilia no change in macrophage numbers	[49]
**rBCG::hIL-18**	human DCs	**allergic asthma**—induction of IL-10 and IP-10 lack of IL-23 in the presence of Der p 1 less IL-5 was produced by naive T cells	[51]
**rBCG::IL-15**	C57BL/6	**bladder cancer** prolonged survival of mice suppression of tumor growth increased infiltration of neutrophiles and expression of the chemokines MIP-2 and MIP-1α	[57]
**rBCG::MUC1-GM-CSF**	CB17, SCID	**breast cancer** limited tumor growth and prolonged survival of mice enhanced IFN-γ production and CD8^+^ T cell response	[67]
**rBCG::hGM-CSF-BZLF1**	C57BL/6	**cancer** delayed tumor formation prolonged mouse survival through the induction of cytotoxic T lymphocytes.	[68]
**rBCG::hGM-CSF-LMP2A**	C57BL/6	**cancer** Induction of cytotoxic T cell responses able to lyse EBV^+^ tumor cells slower tumor growth decreased tumor size and weight	[69]
**rBCG::IL-35**	BALB/c	**RSV-induced asthmatic newborn mouse model** limited lung tissue injury and the deposition of collagen fibers reduced serum IgE levels reduced numbers of inflammatory cells in bronchoalveolar lavage fluids of asthmatic mice inhibited Th17 cell differentiation increased Treg cell numbers	[76]

^a^ Abbreviations: DC, dendritic cell; DTH, delayed type hypersensitivity; EBV, Epstein Barr virus; GFP, green fluorescent protein; OVA, ovalbumin; PBMC, peripheral blood mononuclear cells; PPD, purified protein derivative; RSV, respiratory syncytial virus; Tg, transgenic.

## 3. Recombinant BCG Strains Producing Bacterial Effectors

### 3.1. ADP-Ribosylating Toxins

In addition to cytokines, rBCG-producing bacterial immune modifiers have also been constructed (Table 2). Bacterial toxins are well known for their immune-modulatory functions and adjuvant properties. As such, the cholera toxin family, including *Escherichia coli* heat-labile toxins (LT), are among the most potent mucosal adjuvants [78]. These toxins are members of the ADP-ribosylating toxin family, composed of an enzymatically active A subunit and a pentameric B oligomer responsible for toxin binding to target-cell receptors. Both the A subunit and the B oligomer display immune-modulatory functions. In addition to its mucosal adjuvant activity, the non-toxic cholera toxin B subunit (CTB) has also been proposed for the treatment of inflammatory diseases [79].

In an effort to augment BCG immunogenicity when delivered mucosally, both CTB [80] and the B subunit of LT (LTB) [81] have been produced by rBCG strains. By the use of three different expression vectors, LTB was produced at three different sub-cellular locations of rBCG: intracellular, cell-wall associated and secreted into the extracellular milieu [81]. Intraperitoneal immunization with rBCG::LTB producing cell-wall-associated LTB resulted in a dose-dependent increase of serum anti-LT IgG and IgA. While the IgG response waned rapidly, the IgA responses persisted for up to 168 days after vaccination. The response induced by this strain was stronger than those induced by the other rBCG::LTB strains. The IgG response was characterized by a predominance of IgG2a over IgG1, suggesting a Th1 type response, while immunization with LTB induced predominantly IgG1. Oral immunization with any of the rBCG::LTB strains did not result in detectable IgG to LTB, even after repeated high-dose delivery. In contrast, IgA responses were detected after oral immunization with strains producing cell-wall-associated or secreted LTB. Anti-LTB IgA was also detected in the stool samples, especially of mice orally immunized with rBCG::LTB.

The adjuvant activity of LT produced by rBCG was not evaluated in this study, but was addressed in a study using an rBCG strain that produces CTB [80]. This rBCG::CTB strain produced cell-associated and secreted CTB assembled into its pentameric form, as evidenced by its ability to bind the ganglioside GM1, which is essential for the expression of its mucosal adjuvant activity [82]. Intranasal immunization with rBCG::CTB resulted in higher BCG-specific IgA responses with longer persistence in the bronchoalveolar lavage fluids than immunization with pBCG. Systemic IgG responses were also increased after rBCG::CTB vaccination, either given intranasally or intraperitoneally, compared to immunization with the parental strain. The improved mucosal IgA response after intranasal rBCG::CTB vaccination was paralleled with increased induction of TGF-ß1 in the bronchoalveolar lavage fluid, a cytokine involved in CTB-induced IgA switching [83] but also in mucosal tolerance [84]. When ovalbumin was given intraperitoneally together with intranasal BCG, both BCG strains increased ovalbumin-specific mucosal IgA responses, but the response was faster and stronger when rBCG::CTB was given intranasally instead of pBCG. Although these studies provide proof of concept, at least in mice, that production of CTB by rBCG may be an attractive option to enhance mucosal immunity, safety issues, especially Bell’s Palsy, linked to nasal CTB or LTB [85] have so far limited applications in humans.

More recently, the genetically detoxified A subunit of LT (LTA) was also produced by rBCG in an attempt to improve the protective efficacy of BCG against tuberculosis through the Th1-enhancing effect of LT [86]. Genetically inactivated LTA produced by rBCG::LTA significantly improved the protective effect of BCG against *M. tuberculosis* challenge in mouse models, as assessed by reduced *M. tuberculosis* load in the lungs and reduced lung pathology in subcutaneously vaccinated mice. This was paralleled by increased systemic IFN-γ and TNF-α responses in rBCG::LTA- compared to pBCG-vaccinated mice. Curiously, low-level production of LTA by rBCG::LTA resulted in stronger protection and higher IFN-γ, IL-17 and IL-6 production than high-level LTA production before challenge. However, this was no longer the case after *M. tuberculosis* challenge and was hypothesized to be due to the induction of TGF-ß by the rBCG::LTA followed by challenge infection. Importantly, low-level LTA-producing rBCG::LTA also protected mice against a very high challenge dose of *M. tuberculosis* or with a highly virulent *M. tuberculosis* Beijing strain, while pBCG did not.

In a follow-up study, the effect of the same rBCG::LTA strain on innate immunity was compared to that of the parental strain [87]. Intraperitoneal administration of rBCG::LTA induced stronger NO production by peritoneal cells compared to pBCG, while H_2_O_2_ production was similar. However, NO production was not sustained, whereas H_2_O_2_ production persisted only in the rBCG::LTA-vaccinated animals. Peritoneal lymphocyte numbers were also increased in rBCG::LTA- compared to pBCG-vaccinated mice, and this was evident both for CD4^+^ and for CD8^+^ T cells. Subcutaneous immunization also revealed that rBCG::LTA induced significantly longer IFN-γ, TNF-α, IL-6, IL-17 and IL-10 production by lung cells than pBCG. Finally, after *M. tuberculosis* challenge, rBCG::LTA-vaccinated mice had stronger TNF-α^+^ and bifunctional IL-2^+^TNF-α^+^CD4^+^ T cell responses than pBCG-vaccinated animals.

Pertussis toxin (PT) is another member of the ADP-ribosylating toxin family. It is composed of five different subunits organized in an A-B structure, in which the A subunit (PTA), also named S1, carries the enzymatic ADP-ribosyltransferase activity, and the B moiety, composed of subunits S2 to S5, is responsible for receptor binding (for a recent review, see [88]). Initially engineered as a vaccine candidate to simultaneously protect against tuberculosis and pertussis [89,90], rBCG::PTA producing an enzymatically inactive PTA subunit was later found to induce more IL-8, IL-6 and IL-10 by blood cells from human donors than the parental strain [91]. PBMCs from human donors exposed to rBCG::PTA also expressed higher cytotoxic activity against cancer cells than when exposed to pBCG. This was measured by using a human bladder cancer cell line and thereby confirmed previous non-clinical studies in a mouse model of bladder cancer [92]. Periurethral administration of the rBCG::PTA strain to mice induced higher levels of TNF-α and IL-10 in the bladder of mice than administration of pBCG. This difference was also observed in mice challenged with bladder tumor cells followed by BCG treatment and resulted in significant differences in bladder weight reduction. A follow-up study confirmed the superiority of rBCG::PTA over pBCG treatment in the reduction of bladder weight, as well as increased TNF-α and IL-10 expression in the bladder of mice having received bladder cancer cell instillation [93]. This study also showed stronger cytotoxic activity of spleen cells towards tumor cells and improved mouse survival time in the rBCG::PTA-treated versus the pBCG-treated animals.

Very recently, rBCG::PTA was also found to be a stronger inducer of innate immune memory than pBCG [94]. Mouse bone-marrow-derived macrophages and peritoneal macrophages stimulated with rBCG::PTA produced more inflammatory cytokines, such as TNF-α and IL-6, upon heterologous challenge with *Candida albicans*, *Staphylococcus aureus* or lipopolysaccharide than pBCG-stimulated cells. Enhanced trained innate immunity by rBCG::PTA compared to pBCG was also reflected by increased glucose consumption and lactose production in rBCG::PTA-treated cells. In vivo, subcutaneous rBCG::PTA administration to C57BL/6 mice four weeks prior to lipopolysaccharide challenge led to substantially increased serum IFN-γ levels four hours after challenge compared to pBCG priming. In SCID mice, rBCG::PTA vaccination significantly reduced the fungal burden in the kidney upon *C. albicans* challenge, unlike vaccination with pBCG.

BCG has also been evaluated for the prevention or treatment of allergic asthma, albeit with conflicting results [95]. Although the anti-asthma effect was well evident in mouse models, it was less clear in humans, which provides room for improvement. Given the immunomodulatory ability and Th1 skewing of rBCG::PTA, this strain was compared to pBCG for its effect on experimental asthma in a mouse model sensitized and challenged with Ovalbumin [96]. Intranasal administration of rBCG::PTA resulted in decreased eosinophilia in the lungs of the allergic mice to a significantly higher degree than in the lungs of pBCG-treated animals. This protective effect was also reflected by the measurements of allergic airway hyper-reactivity upon ß-methacholine challenge, IL-5 and IL-13 levels in the lungs and lung pathology, for which rBCG::PTA vaccination performed better than pBCG vaccination. rBCG::PTA vaccination also resulted in higher numbers of IFN-γ-secreting cells and lower numbers of IL-4-secreting cells in the lungs of allergic mice than pBCG vaccination, and the IL-12-IFN-γ axis was found to be key in the protective effect of rBCG::PTA, as investigated by the use of IL-12-deficient mice, in which the protective effects of the rBCG::PTA against eosinophilia and Th2 cytokine secretion in the lungs were partially reduced.

The mechanism by which LTA and PTA enhance BCG immunogenicity and Th1 skewing is not yet known. As the modulatory effects of the respective rBCG strains were demonstrated with enzymatically inactivated LTA and PTA derivatives, the mechanism is independent of the ADP-ribosyltransferase activity of the two toxin subunits. Nevertheless, they may perhaps still interact with their cognate Gα protein ADP-ribose acceptor substrates and potentially modify their function in G-protein-coupled receptor signal transmission. Further studies are needed to identify the mode of action of enzymatically inactivated A subunits of bacterial ADP-ribosylating toxins.

### 3.2. Pore-Forming Bacterial Toxins

In addition to ADP-ribosylating toxin subunits, rBCG strains producing pore-forming toxins, such as listeriolysin O and perfringolysin, have also been generated and tested for their immune-modifying properties.

### 3.3. Listeriolysin O

Listeriolysin O, a major virulence factor of *L. monocytogenes*, mediates the escape of the organism from the host–cell vacuoles to the cytosol without causing cell death, which is thought to lead to listerial antigen presentation via MHC class I molecules and to subsequent activation of CD8^+^ T cells (for review, see [97]). BCG induces strong CD4^+^ T cell responses but relatively weak CD8^+^ T cell responses in contrast to *M. tuberculosis*, which induces both strong CD4^+^ and strong CD8^+^ responses, which may be important for protective immunity to tuberculosis [98]. In order to improve MHC class I presentation by BCG, recombinant strains secreting listeriolysin (rBCG::*hly*) were therefore constructed and shown to induce MHC class I presentation against co-phagocytosed ovalbumin by IFN-γ-activated murine macrophage-like cells, as well as ovalbumin-specific CD8^+^ T cell activation [99]. Human DCs cultured with rBCG::*hly* were also shown to be more effective in inducing cytotoxic T cells than DCs cultured with pBCG [100].

In subsequent studies, a similar rBCG strain was therefore tested for protection against tuberculosis in a mouse model [101]. In this rBCG version named rBCGΔ*ureC::hly*, the *ureC* gene was removed in order to decrease the phagosomal pH for optimal listeriolysin activity. Intraperitoneal vaccination with rBCGΔ*ureC::hly* resulted in superior protection than vaccination with the parental strain in a mouse model of aerosol infection with *M. tuberculosis*, as evidenced by the decreased bacterial burden in the lungs early, and especially late, after challenge infection. Importantly, rBCGΔ*ureC::hly* also protected against a highly virulent *M. tuberculosis* Beijing/W strain, against which the parental BCG strain provided no significant protection. Interestingly, rBCGΔ*ureC::hly* was also safer in an SCID mouse model. While high doses of pBCG given intravenously killed SCID mice within 4 weeks after administration, survival of rBCGΔ*ureC::hly*-infected SCID mice was prolonged up to 11 weeks. It also did not persist as long in the draining lymph nodes and in the spleen of subcutaneously vaccinated C57BL/6 mice than pBCG and did not disseminate to the lungs [102]. The improved protective efficacy of rBCGΔ*ureC::hly* compared to pBCG could be linked to enhanced release of mycobacterial antigens into the cytosol of infected macrophages, resulting in profound apoptosis of the macrophages, which together increased MHC class I antigen loading and therefore presumably enhanced CD8^+^ T cell activation.

rBCGΔ*ureC::hly* also induced stronger CD4^+^ T cell responses in mice than pBCG. Following *M. tuberculosis* infection, cytokine production by CD4^+^ T cells, including IFN-γ, IL-17, IL-6, GM-CSF and IL-2, but not Th2 cytokines, was increased by lung cells from rBCGΔ*ureC::hly*-vaccinated mice stimulated with PPD compared to the cells from pBCG-vaccinated animals [103]. Interestingly, only rBCGΔ*ureC::hly* vaccination induced a strong Th17 type response, together with a Th1 response in spleen cells, while pBCG only induced Th1 cytokines by spleen cells. Polyfunctional CD4^+^ T cells producing IL-2 and TNF-α, or producing IL-2, IFN-γ and TNF-α, or IL-2, IFN-γ, TNF-α and IL-17 were also preferentially induced by rBCGΔ*ureC::hly*, as were IL-17-producing γδ T cells. This improved IL-17 production after rBCGΔ*ureC::hly* vaccination may be linked to the induction of the pathogen-associated molecular pattern-activated intracellular signaling cascade via the release of cytosolic mycobacterial antigens due to the activity of listeriolysin. It may also be due to the induction of apoptosis by antigen-presenting cells in rBCGΔ*ureC::hly*-vaccinated mice and/or the activation of inflammasomes.

Inflammasome activation has indeed been shown to occur upon rBCGΔ*ureC::hly* vaccination [104]. Mycobacterial DNA was found to be released in rBCGΔ*ureC::hly*-infected, but not pBCG-infected, macrophage-like cell lines, which activated the inflammasome via cytosolic DNA sensing by its absence in melanoma 2 (AIM2). This led to caspase 1, 3 and 7 activation and release of pro-inflammatory IL-1ß. It also led to the induction of autophagy, which may promote antigen presentation to T cells. This was also evidenced in vivo, as rBCGΔ*ureC::hly* vaccination of mice resulted in strong upregulation of genes associated with innate immunity, including *Il-18*, *Il-1ß* and *Myd88*.

rBCGΔ*ureC::hly* also induces antigen-specific central memory CD4^+^ T cells, which are likely to be important for sustained immune protection [102]. Both rBCGΔ*ureC::hly* and pBCG induced comparable numbers of antigen-specific CCR7^−^CXCR5^−^PD-1^−^ effector memory T cells, but only rBCGΔ*ureC::hly* vaccination induced CCR7^+^CXCR5^+^PD-1^+^ central memory T cells in addition. Both memory T cell subsets persisted after the clearance of rBCGΔ*ureC::hly*. Adoptive transfer or these central memory T cells induced by rBCGΔ*ureC::hly* vaccination resulted in protection against *M. tuberculosis* challenge, while transfer of effector memory T cells did not, illustrating the role of the former subset in the improved protective effect of the recombinant strain.

Under the name of VPM1002, the rBCGΔ*ureC::hly* vaccine candidate is now in clinical development and has already shown safety and immunogenicity in healthy adult volunteers [105], as well as in HIV-uninfected newborns [106]. In adults without previous BCG vaccination, rBCG induced a stronger IFN-γ response to PPD than pBCG, when measured 180 days after vaccination, while these responses were similar in BCG-vaccinated individuals [105]. IFN-γ-producing CD4^+^ T cell frequencies were also similar between the VPM1002 and the pBCG groups, while, interestingly, the frequencies of IFN-γ-producing CD8^+^ T cells were higher in the VPM1002 than in the pBCG group. Antibodies against PPD were also more frequently induced in VPM1002-vaccinated subjects than in pBCG-vaccinated subjects. In HIV-negative newborns VPM1002 also induced more IL-17^+^CD8^+^ T cells than pBCG [106].

To further increase the immunogenicity and protective efficacy of rBCGΔ*ureC::hly* by enhancing apoptosis, the anti-apoptotic *nuoG* gene encoding an NADH dehydrogenase 1 subunit was deleted from the chromosome of rBCGΔ*ureC::hly* [107]. This deletion resulted in a further increase in the induction of lymph node CD4^+^ follicular helper T cell and effector memory T cell frequencies over the original rBCGΔ*ureC::hly* strain, as well as in the enhanced expression of *Ifng* and IFN-γ-induced genes. The deletion of *nuoG* also improved the protective efficacy of rBCGΔ*ureC::hly*, especially when the bacterial burden in the lungs was evaluated at late time points after *M. tuberculosis* infection.

Given the enhanced innate immune induction by rBCGΔ*ureC::hly* and its rapid clearance compared to pBCG, rBCGΔ*ureC::hly* has also been proposed for the treatment of NMIBC. In a phase I clinical trial with VPM1002BC, the formulation for bladder cancer indication has recently been completed in patients with a previous failure of the conventional BCG treatment [108]. In this study, six male patients with a high-grade urothelial carcinoma < T2, who had relapsed after previous BCG treatment and did not undergo cystectomy, received six weekly instillations of VPM1002BC, followed by one year of three weekly instillations 3, 6 and 12 months later. The treatment was generally well-tolerated, and no bacteria were shed after 24 h, which indicates faster clearance than after conventional BCG treatment. Urine IL-2, IFN-γ, TNF-α, IL-17, GM-CSF and IL-13, as well as plasma IL-2, IFN-γ and TNF-α, levels increased after treatment. PPD-specific GM-CSF^+^Ki67^+^CD4^+^ T cells and IFN-γ^+^CD4^+^ T cells were also induced in the blood over the course of treatment. Whether this results in favorable clinical outcomes remains to be established in future studies.

### 3.4. Perfringolysin O

Perfringolysin O is another bacterial pore-forming toxin used to improve BCG efficacy against tuberculosis. Unlike listeriolysin O, produced by an intracellular pathogen, perfringolysin O is produced by the extracellular pathogen *Clostridium perfringens* and is toxic to cells when produced by intracellular organisms [109]. Therefore, a mutant version of this toxin had to be used to prepare the rBCG strain, named AERAS-401 [110]. This toxin analog, in which glycine at position 137 was replaced by glutamine, was active at neutral and acidic pH and had a shortened half-life [109]. AERAS-401 secreted active perfringolysin O into the culture medium, as assessed by the hemolytic activity on sheep red blood cells, but the vaccine strain lacked cytotoxic activity upon infection of macrophage-like cells [110]. AERAS-401-infected SCID mice survived longer than pBCG-infected SCID mice, suggesting an improved safety profile. However, immunogenicity and protective efficacy in mice were not studied with AERAS-401. Instead, this strain was used as a host strain to overproduce mycobacterial antigens, such as Ag85A and Ag85B, and the resulting recombinant strain was found to induce stronger IFN-γ responses in mice, as well as a somewhat longer survival upon *M. tuberculosis* challenge than pBCG. However, the bacterial burden of the challenged mice in the lungs and spleen were similar in rBCG- or pBCG-vaccinated animals.

Both strains also provided similar levels of protection against a low-dose aerosol challenge infection of mice and guinea pigs with a variety of recent clinical isolates of *M. tuberculosis*, although rBCG performed slightly better against a clinical isolate from Western Cape in the guinea pig model [111]. However, long-term protection against *M. tuberculosis*-induced death in guinea pigs infected with clinical isolates was weaker after rBCG vaccination than after pBCG vaccination.

In rhesus macaques, priming with this rBCG vaccine followed by boosting with a recombinant non-replicating adenovirus coding for Ag85A and Ag85B resulted in higher IFN-γ responses to these two antigens than priming with pBCG followed by the same adenovirus vaccine candidate [112]. Curiously, IFN-γ responses to BCG extracts or PPD were lower in the rBCG-primed than in the pBCG-primed primates. Compared to pBCG, vaccination with rBCG also induced stronger CD4^+^ and stronger CD8αα^+^ memory T cell proliferation after stimulation with Ag85B, but not after stimulation with Ag85A, PPD or BCG extracts. Polyfunctional antigen-specific CD4^+^ and CD8αα^+^ T cells simultaneously producing TNF-α, IFN-γ and IL-2 were also induced by rBCG vaccination in some animals, but they were also induced by pBCG vaccination. Unfortunately, these studies do not inform whether the differences in immune responses induced by the two BCG strains are due to the increased production of Ag85A and Ag85B, or the presence of perfringolysin in the rBCG strain. Interestingly, though, safety studies in SCID mice comparing this rBCG strain with AERAS-401 showed that the overproduction of the mycobacterial antigens further improved survival of the rBCG strain [113]. The reasons for this difference have not yet been elucidated.

Despite the at-best modest improvement of rBCG over pBCG documented in pre-clinical studies, the rBCG strain proceeded to clinical development. A phase I study involving 24 adults and comparing pBCG to a low and a high dose of rBCG showed that only the high dose rBCG induced T cell responses to Ag85A and Ag85B, while all three vaccination groups developed comparable T cell responses to BCG [114]. However, in the high dose rBCG group, whole blood incubated with BCG led to significant inhibition of BCG growth, which was not seen for the other treatment groups. Surprisingly, this was negatively correlated with IL-1ß and monocyte chemokine expression but correlated positively with the early induction of NK and cytotoxicity-associated genes measured in PBMCs.

Unexpectedly, rBCG vaccination caused VZV reactivation in two subjects vaccinated with the high dose rBCG, which resolved after treatment. Both patients had an HLA allele that had previously been associated with anti-IFN-γ auto-immunity. However, analysis of the plasma prior to and after vaccination did not reveal elevated levels of IFN-γ-neutralizing activity, while the PBMCs from both patients produced much more IFN-γ upon BCG stimulation than the PBMCs from the other vaccinees in the high-dose group. This safety signal led to the discontinuation of rBCG clinical development.

Interestingly, even in the absence of perfringolysin, overproduction of Ag85B in rBCG::Ag85B resulted in enhanced in vitro human bladder cancer cell growth inhibition compared to pBCG [115]. This enhanced cytotoxic activity was linked to a decreased expression of the anti-apoptotic *bcl-2* and *survivin* genes, and an increase in the pro-apoptotic *bax* gene expression. The genes coding for apoptosis-inducing factor, endonuclease G, caspase-8, caspase-9 and caspase-3 were also significantly upregulated in rBCG::Ag85B-treated cells compared to pBCG-treated cells. The reasons why overproduction of Ag85B results in enhanced apoptosis induction of BCG are not completely elucidated but may be related to the fibronectin-binding activity of Ag85B, which may result in increased mycobacterial adherence to the tumor cells.

In a different study, it was found that over-production of Ag85B in rBCG::Ag85B induces aggresomes in antigen-presenting cells (APC), which triggers autophagy, which in turn directs antigens to lysosomes, facilitating their degradation and presentation by MHC molecules, thereby increasing T cell immune responses [116]. An rBGC::85B strain overproducing a specific peptide from the *M. tuberculosis* protein CFP-10 further increased autophagy, elicited a strong reactive oxygen species (ROS) response in cord blood-derived macrophages through TLR2 signaling and enhanced antigen presentation by DCs [117]. T cell immunogenicity was also stronger in mice vaccinated with this strain compared to the strain that only over-produced Ag85B with a stronger induction of effector and central memory T cells as well as multifunctional T cells and functional CD8^+^ T cells, which translated to improved protection against *M. tuberculosis* challenge in mice.

## 4. Recombinant BCG Strains Producing Other Non-Bacterial Immunomodulatory Proteins

Enhancement of BCG-induced immune responses has also been demonstrated for rBCG strains producing other molecules that target the immune system, such as a variety of ligands of innate immune cells (Table 2). The Fms-like tyrosine kinase 3 ligand (Ftl3L) contributes to the development of APCs, including DCs, and was therefore over-produced in an rBCG strain [118]. Vaccination of mice with rBCG::Flt3L indeed induced stronger IFN-γ responses in the spleen and draining lymph nodes compared to pBCG at early time points after administration. However, this did not lead to improved protection by rBCG:Flt3L over pBCG in a murine *M. tuberculosis* aerosol challenge model. Interestingly, though, the recombinant strain proved to be safer than the parental strain in immune-deficient mice.

Similar observations were made with an rBCG strain producing and secreting monocyte chemotactic protein 3 (MCP-3) [119]. MCP-3 is an agonist of several chemokine receptors on a variety of innate and adaptive immune cells. Functional MCP-3 could be produced and secreted by rBCG::MCP-3, as shown by increased numbers of CD4^+^ and of CD8^+^ T cells in the peritoneal exudate upon intraperitoneal infection with this strain compared to mice that had received pBCG. Subcutaneous immunization with rBCG::MCP-3 also induced an increase in the number of BCG-specific IFN-γ-secreting cells compared to vaccination with pBCG. However, as with the rBCG::Flt3L-vaccinated mice, the rBCG::MCP-3-vaccinated mice did not afford improved protection against *M. tuberculosis* challenge, similar to rBCG::Flt3L. rBCG::MCP-3 was also safer in RAG^−/−^ mice, which survived significantly longer after infection with these rBCG strains than with pBCG.

Th1 responses were also augmented by the production of the pro-apoptotic protein BAX in rBCG, as this protein triggers cytochrome c release from the mitochondria, which induces apoptosis in macrophages via the mitochondrial pathway [120]. The approach was based on the fact *M. tuberculosis* inhibits apoptosis of macrophages in an attempt to escape the host immune response. Therefore, infection with rBCG::BAX was expected to enhance apoptosis over pBCG and thus increase immune responses. IFN-γ, TNF-α and IL-2 responses to Ag84B were indeed stronger after rBCG::BAX than after pBCG vaccination, whereas IL-4 responses to Ag85B were deceased, indicating a further shift towards a Th1 profile induced by rBCG::BAX over pBCG. Curiously, this was not seen when PPD was used in the assay instead of Ag85B. Strangely, the enhanced Th1 shift was also observed after vaccination with rBCG that only contains the vector sequences without the *BAX* gene. However, this finding was not discussed in the article by Li et al. [120]. Therefore, it remains unclear what the actual contribution of BAX was in the observed immune responses. Furthermore, the rBCG stains were not compared to pBCG for their protective potential against tuberculosis or other diseases.

Instead of manipulating macrophage apoptosis, rBCG can also be engineered to counteract negative regulators of immune responses, such as the suppressor of cytokine signaling 1 (SOCS1), which is a negative regulator of JAK/STAT signaling, important for the induction of antimicrobial NO [121]. Mycobacteria strongly induce SOCS1 expression, which results in impaired IFN-γ secretion by macrophages, thereby negatively affecting intracellular mycobacterial control [122]. In macrophages infected with an rBCG strain that produces a SOCS1 antagonist (rBCG::SOCS1DN), higher STAT1 phosphorylation levels were found compared to cells infected with the control strain [121]. The expression of the gene encoding iNOS was also higher in rBCG::SOCS1DN-infected than in pBCG-infected cells. Although the production of SOCS1DN by rBCG::SOCS1DN did not affect bacterial growth in vitro, it resulted in accelerated clearance in mice after intratracheal infection, while lung inflammation persisted longer than after infection with pBCG, potentially due to higher NO production by the recombinant strain. rBCG::SOCS1DN also increased the surface expression of CD80, CD86 and MHC molecules, in addition to proinflammatory cytokine secretion by DCs compared to pBCG, resulting in augmented IFN-γ, IL-13 and IL-17 production by CD4^+^ T cells, as well as augmented IFN-γ production by CD8^+^ T cells [123]. In mice, rBCG::SOCS1DN vaccination resulted in significantly larger numbers of central memory and effector memory CD8^+^ T cells and significantly improved protection against *M. tuberculosis* infection compared to pBCG vaccination.

Recombinant cathepsin S has also been produced by rBCG strains [124]. Cathepsin S is a protease involved in antigen presentation by MHC class II molecules, and the expression of its gene is repressed by BCG which diminishes its immunogenicity [125]. Therefore, infection with rBCG::CatS producing cathepsin S was expected to restore cathepsin S function and enhance MCH class II antigen presentation. In contrast to pBCG, infection of macrophages with rBCG::CatS led to phagosomal maturation, rapid killing of the mycobacteria, elevated expression of mature MHC class II molecules at their surface and ensuing increased presentation of Ag85B epitopes to T cells [124]. The in vivo effect of rBCG::CatS on protection against *M. tuberculosis* infection was not examined in this study.

## 5. Recombinant BCG Strains Producing Non-Protein Immunomodulatory Molecules

The early interactions of BCG with phagocytes could also be modulated by the expression of heterologous non-protein immune modulators (Table 2). Phenolic glycolipid 1 (PGL-1) is a compound that is located at the outer surface of *Mycobacterium leprae* and is not produced by BCG. An rBGC strain containing the *M. leprae* genes required for the biosynthesis of the trisaccharide domain of PGL-1 altered the route of mycobacterial entry into human macrophages and DCs [126]. The rBCG::PGL-1 strain invaded macrophages and DCs via complement receptor 3 significantly more efficiently and grew better in these cells than pBCG. However, the expression of PGL-1 impaired the BCG-induced NF-κB signaling pathway in monocytes, leading to a lowered production of TNF-α. However, in a follow-up study, TNF-α production by human monocytes was higher after infection with rBCG::PGL-1 than with pBCG, as was the production of IL-12p70 [127], and pre-exposure of the monocytes with rBCG::PGL-1 primed the cells for increased production of IL-1ß and IL-10 by LPS but blocked the induction of VEGF by BCG [127]. The reasons for these conflicting observations are not well understood but may be due to the different cell types used in the two studies (macrophages versus monocytes).

Unlike rBCG::PGL-1, the rBCG strain engineered to produce elevated levels of cyclic di-AMP (c-di-AMP), a STING agonist, showed a clear enhanced adjuvant effect over pBCG [128]. The production of c-di-AMP is catalyzed by bacterial diadenylate cyclases using ATP as the substrate. In *M. tuberculosis* the enzyme is encoded by *rv3686* (*disA*) [129], but expression levels of *disA* are low in both *M. tuberculosis* and BCG [128]. Overexpression of *disA* in BCG resulted in a more than three-fold increased production of c-di-AMP, and vaccination of rBCG::DisA resulted in a stronger IFN-γ response in the lungs of mice than vaccination with pBCG, while other cytokines, such as IL-2 and IL-10, were induced at similar levels by the two BCG strains. Most strikingly, however, after infection with *M. tuberculosis*, the rBCG::DisA-vaccinated mice had a significantly stronger IFN-γ, IL-2 and IL-10 response in the lungs and spleen than pBCG-vaccinated mice. However, these differences did not lead to improved protection by rBCG::DisA against *M. tuberculosis* infection over pBCG vaccination.

In another study with a different rBCG::DisA strain in which *disA* expression is under the control of the strong *hsp60* promoter, improved protection over pBCG against *M. tuberculosis* challenge could be demonstrated using a guinea pig model [130]. In this strain, *disA* expression was increased 300-fold and c-di-AMP production 15-fold compared to the parental strain [131]. This overexpression resulted in increased induction of IRF3 and *ifnb* expression in infected macrophages and subsequent increased levels of TNF-α, IL-6 and IL-1ß secretion [130]. Compared to pBCG, vaccination of guinea pigs with rBCG::DisA also reduced lung pathology and bacterial burden in the lungs and spleen upon infection with *M. tuberculosis*.

Recently, this rBCG::DisA strain was also tested for its efficacy in a rat model of NMIBC [131]. After induction of bladder cancer by treatment with *N*-methyl-*N*-nitrosourea, the rats received six weekly doses of either rBCG::DisA or pBCG. Histological analyses revealed that rBCG::DisA-treated rats had significantly lower pathologic grades, tumor involvement indexes and highest tumor stages compared to animals treated with pBCG. This was paralleled with increased pro-inflammatory transcriptional signatures and pro-inflammatory macrophages but decreased immunosuppressive macrophages in the rBCG::DisA compared to the pBCG group. Similar observations were made in a murine heterotopic, syngeneic urothelial cancer model. This improved efficacy could be related to the stronger innate immune training by rBCG::DisA compared to pBCG. Macrophages incubated with rBCG::DisA induced stronger expansion of TNF-α-producing CD45^+^CD11b^+^F4/80^+^ cells and lower expression of the immunosuppressive surface markers CD206 and CD124 compared with pBCG. rBCG::DisA-exposed human macrophages were also more phagocytic than macrophages exposed to pBCG and underwent stronger epigenetic and metabolic re-programing towards pro-inflammatory signatures.

**Table 2 vaccines-10-00827-t002:** Recombinant BCG strains producing bacterial and other non-bacterial immunomodulatory effectors.

Name of rBCG	Name of Effector	Experimental Model	Main Findings	Reference
**rBCG Strains Producing Bacterial Effectors**				
**rBCG::LTB**	*E. coli* heat labile toxin B subunit (LTB ^a^)	BALB/c intraperitoneal BALB/c oral	dose-dependent increase of serum anti-LT IgG and IgA predominance of IgG1 non-detectable IgG detection of anti-LTB IgA	[81]
**rBCG::CTB**	Cholera toxin B subunit (CTB)	BALB/c intranasal BALB/c intranasal/intraperitoneal BALB/c intranasal	higher BCG-specific IgA responses longer persistence in the bronchoalveolar lavage increased systemic IgG responses improved mucosal IgA response increased induction of TGF-ß1 in the bronchoalveolar lavage fluid	[80,83]
**rBCG::LTA**	*E. coli* heat labile toxin A subunit (LTA)	BALB/c subcutaneous BALB/c intraperitoneal	improved the protective effect against *M. tuberculosis* by reduced bacterial load in the lungs and reduced lung pathology increased IFN-γ and TNF-α responses low level production of LTA by rBCG::LTA induced stronger protection, higher IFN-γ, IL-17, IL-6 than high level production of LTA stronger NO production increase of the peritoneal lymphocyte numbers longer lasting IFN-γ, TNF-α, IL-6, IL-17, IL-10 production by lung cells stronger TNF-α and IL-2^+^TNF-α^+^CD4^+^ T cell response after *M. tuberculosis* challenge	[86,87]
**rBCG::PTA**	Pertussis toxin A subunit (PTA)	human PBMC/ human bladder cancer cell line mice, periurethral SCID mice, subcutaneous BALB/c intranasal/mouse model of asthma	increased production of IL-8, IL-6 and IL-10 higher cytotoxic activity against bladder cancer cells higher levels of TNF-α and IL-10 in the bladder of mice reduction of bladder weight, increased TNF-α and IL-10 expression in the bladder of mice stronger cytotoxic activity of spleen cells towards tumor cells improved mouse survival time stronger inducer of innate immune memory increased TNF-α and IL-6 by murine macrophages increased serum IFN-γ levels reduced the fungal burden in the kidney upon *C. albicans* challenge decreased eosinophilia higher numbers of IFN-γ-secreting and lower numbers of IL-4-secreting cells in the lungs IL-12-IFN-γ axis was found to be key in the protective effect of rBCG::PTA	[91,92,93,94,96]
**rBCG::*hly*** **rBCGΔ*ureC::hly***	**listeriolysin O**	murine macrophage-like cell lines human DCs BALB/c intraperitoneal C57BL/6 intravenous BALB/c subcutaneous	induce MHC class I presentation more effective in inducing cytotoxic T cells superior protection against *M. tuberculosis* decreased bacterial burden in the lungs safer in SCID mice prolonged survival of mice induction of antigen-specific central memory CD4^+^ T cells stronger CD4^+^ T cell responses increased production of IFN-γ, IL-17, IL-6, GM-CSF and IL-2 by lung cells stronger Th17 type response inflammasome activation induction of autophagy strong upregulation of *Il-18*, *Il-1ß* and *Myd88* enhanced apoptosis	[99,100,101,102,103,104,107]
	VPM1002 VPM1002BC,	healthy volunteers HIV-uninfected newborns bladder cancer patients	strong IFN-γ response to PPD higher frequencies of IFN-γ-producing CD8^+^ T cells more frequently induced antibodies against PPD increased numbers of IL-17^+^CD8^+^ T cellsfast clearance of bacteria urine IL-2, IFN-γ, TNF-α, IL-17, GM-CSF and IL-13, as well as plasma IL-2, IFN-γ and TNF-α levels increased after treatment PPD-specific GM-CSF^+^Ki67^+^CD4^+^ T cells and IFN-γ^+^CD4^+^ T cells induction	[105,106,108]
**AERAS-401**	Perfringolysin O	SCID subcutaneous	longer survival of mice overproduction of the mycobacterial antigens improved survival of rBCG strains high dose rBCG induced T cell responses to Ag85A and Ag85B	[110,111,112]
**rBCG::Ag85B**		bladder carcinoma cell line 5637	enhanced presentation by DCs increased T cell immunogenicity increased induction of effector and central memory T cells improved protection against *M. tuberculosis* in mice	[115]
**Recombinant BCG Strains Producing Other Non-bacterial Immunomodulatory Proteins**				
rBCG::Flt3L	Fms-like tyrosine kinase 3 ligand	C57BL/6 intramuscular	induced stronger IFN-γ responses in spleen and draining lymph nodes lack of improved protection against *M. tuberculosis*	[118]
rBCG::MCP-3	monocyte chemotactic protein 3 (MCP-3)	C57BL/6 intraperitoneal subcutaneous	increased numbers of CD4^+^ and of CD8^+^ T cells increased numbers of BCG-specific IFN-γ-secreting cell did not improved protection against *M. tuberculosis.*	[119]
rBCG::BAX	pro-apoptotic protein BAX	C57BL/6 subcutaneous	enhanced apoptosis shift towards a Th1 profile	[120]
rBCG::SOCS1DN	suppressor of cytokine signaling 1 (SOCS1)	C57BL/6 subcutaneous	increased STAT1 phosphorylation levels increased the surface expression of CD80, CD86, MHC augmented IFN-γ, IL-13 and IL-17 production by CD4^+^ and CD8^+^ T cells significantly larger numbers of central memory and effector memory CD8^+^ T cells improved protection against *M. tuberculosis.*	[122,123]
rBCG::CatS	cathepsin S	THP-1 cells	phagosomal maturation rapid killing of the mycobacteria elevated expression of mature MHC class II increased presentation of Ag85B epitopes to T cells	[124]
**Recombinant BCG Strains Producing Non-Protein Immunomodulatory Molecules**				
rBCG::PGL-1	Phenolic glycolipid 1 (PGL-1)	Human monocyte stimulation	impaired the BCG-induced NF-kB signalling pathway leading to lowered production of TNF-α higher TNF-α production increased production of IL-1ß and IL-10 by LPS, but blocked the induction of VEGF	[126,127]
rBCG::DisA	cyclic di-AMP	BALB/c subcutaneous	stronger IFN-γ response in the lungs of mice significantly stronger IFN-γ, IL-2 and IL-10 response in lungs and spleen lack of improved protection against *M. tuberculosis* infection	[128]
		Guinea pigs	improved protection against *M. tuberculosis* increased induction of IRF3 and *ifnb* expression in infected macrophages and increased levels of TNF-α, IL-6 and IL-1ß secretion reduced lung pathology and bacterial burden in the lungs and spleen	[130]
		Bladder cancer rat model	lower pathologic grades, tumor involvement indexes and highest tumor stages increased pro-inflammatory transcriptional signatures and pro-inflammatory macrophages, but decreased immunosuppressive macrophages	[131]
		Urothelial cancer model	stronger expansion of TNF-α-producing CD45^+^CD11b^+^F4/80^+^ cells lower expression of the immunosuppressive surface markers CD206 and CD124	

^a^ Abbreviations: CTB, cholera toxin B subunit; DC, dendritic cell; LTA, *E. coli* heat labile toxin A subunit; MCP, monocyte chemotactic protein; PBMC, peripheral blood mononuclear cells; PGL, phenolic glycolipid; PTA, pertussis toxin A subunit.

## 6. Recombinant BCG Deletion Mutants

Immunomodulatory properties of BCG may also be modified or enhanced by deletion rather than by the insertion of genes (Table 3). Several BCG genes have been associated with immunosuppression. As such, mycobacteria produce several anti-oxidants that suppress host immune responses. One of the factors responsible for antioxidant production is the extracytoplasmic sigma factor SigH. An rBGC strain lacking *sigH*, as well as the secretion channel SecA2 and producing a dominant negative mutant version of the superoxide dismutase SodA, was found to induce stronger T cell responses in C57BL/6 mice than pBCG [132]. After vaccination with the rBCGΔ*sigH* strain, re-stimulated spleen cells contained more Th1 type cytokine-producing CD4^+^ T cells than after vaccination with pBCG, while rBCGΔ*sigH* did not persist as long in the spleen compard to pBCG. In BALB/c mice, rBCGΔ*sigH* induced stronger polyfunctional cytokine-producing CD8^+^ T cells, as well as sustained IL-2 production, than pBCG. rBCGΔ*sigH* was further modified by the introduction of a dominant negative mutant form of glutamine synthetase GlnA1 involved in immune evasion [133]. This strain was also cleared more rapidly from the spleens of C57BL/6 mice than pBCG, and a comparison of the protective effect of this strain with pBCG showed improved protection against *M. tuberculosis* challenge in mice, as evidenced by lower bacterial burdens in the spleen and lungs, as well as decreased lung pathology 8 weeks after challenge. However, protection started to wane 26 weeks after the challenge.

An alternative approach to enhancing BCG immunogenicity by gene deletion was to target the zinc metalloprotease gene *zmp1*, which interferes with inflammasome activation and phagosome maturation [134]. Deletion of *zmp1* from the BCG chromosome allows the bacteria to proceed to the late endosomes of macrophages, and vaccination with rBCGΔ*zmp1* results in stronger mycobacterial antigen presentation by DCs than vaccination with pBCG. This enhanced antigen presentation also led to superior T cell responses, including T cell proliferation and IFN-γ production, as well as increased numbers of mycobacterial antigen-specific IFN-γ^+^CD4^+^ and IFN-γ^+^CD8^+^ T cells. Unlike rBCGΔ*sigHsod* described above, rBCGΔ*zmp1* persisted as long as pBCG in the spleens and livers of mice. When tested in a guinea pig model, rBCGΔ*zmp1* conferred significantly better protection against lung colonization by *M. tuberculosis*, while lung pathology was not significantly different between rBCGΔ*zmp1*-vaccinated and pBCG-vaccinated animals [135]. Importantly, SCID mice infected with rBCGΔ*zmp1* survived longer than SCID mice infected with isogenic BCG, suggesting an improved safety profile of rBCGΔ*zmp1* over pBCG.

Inactivation of the *sapM* locus, coding for a secreted acid phosphatase, also improves protection afforded by BCG [136]. BALB/c mice vaccinated with rBCG*sapM*::Tn contained significantly lower amounts of *M. tuberculosis* bacteria in the lungs 8 weeks after challenge and had significantly reduced mortality than mice vaccinated with pBCG. The two strains induced comparable levels of cytokine responses in the lungs. Mycobacterial uptake by macrophages and DCs, as well as phagosomal maturation, autophagy and ROS induction, were also similar between the two strains, indicating that the mechanism of enhanced protective efficacy by the *sapM* mutation was independent of the inhibition of phagosomal maturation arrest, or the induction of autophagy or ROS. Instead, rBCG*sapM*::Tn induced higher levels of IL-1ß, IL-6 and TNF-α by DCs than pBCG and increased the recruitment and activation of DCs in the draining lymph nodes, while the numbers of mycobacterial antigen-specific CD4^+^ and CD8^+^ T cells producing IFN-γ were lower in the lymph nodes and spleens of rBCG*sapM*::Tn-vaccinated mice at early time points after vaccination, compared to pBCG [137], suggesting faster innate immune control by rBCG*sapM*::Tn. Consistently, fewer bacteria were found in the draining lymph nodes and lungs of rBCG*sapM*::Tn-vaccinated compared to pBCG-vaccinated mice. Since the enhanced protective effect of rBCG*sapM*::Tn is thus mainly based on DC activation and migration, suggesting innate immune control as the main mechanism, the stain may perhaps also be useful for non-tuberculosis indications by expressing superior non-specific effects.

Autophagy could be enhanced by the deletion of *BCG_2432c* from the BCG chromosome [138]. This gene is homologous to the *eis* gene of *M. tuberculosis*, shown to enhance the survival of *M. tuberculosis* within macrophages by inhibiting autophagy and the generation of ROS. The rBCGΔ*2432c* mutant induced more autophagy and apoptosis of THP-1 macrophages and mycobacterial antigen presentation by DCs than pBCG, which increased intracellular killing of the mycobacteria. In C57BL/6 mice rBCGΔ*2432c* vaccination also conferred significantly stronger protection in an *M. tuberculosis* challenge model than pBCG vaccination, as shown by significant differences in bacterial burden in lungs and spleen, as well as in pathology between the two different vaccine treatment groups. However, unlike rBCG*sapM*::Tn, rBCGΔ*2432c* also elicited stronger T cell responses, including Th-1 cytokine-producing CD4^+^ and CD8^+^ T cells, in the spleen and lungs. CD4^+^ memory T cells, both IFN-γ^+^ effector memory and IL-2^+^ central memory T cells, were also induced stronger by rBCGΔ*2432c* than by pBCG.

Other deletions in the BCG chromosome have also led to increased CD8^+^ T cell responses [139]. A transposon-mutant BCG library screen of antigen presentation by a macrophage cell line identified 14 mutated operons that caused enhanced antigen presentation to CD8^+^ T cells. These operons were distributed throughout the BCG chromosome and coded for a wide variety of biological functions. One of the mutants was of particular interest, as a comparison with AERAS-401 (see above) showed that this mutant induced significantly higher CD8^+^ T cell responses and CD8^+^ T cell priming than AERAS-401. However, the transposon was inserted into an operon with an unknown function, and therefore, the molecular mechanism of increased MHC class I presentation remains unknown. It is also not known whether the increase in CD8^+^ T cell responses translates into better protection against *M. tuberculosis* challenge or other diseases.

**Table 3 vaccines-10-00827-t003:** rBCG with chromosomal deletions.

Name of rBCG	Gene Product	Experimental Model	Main Findings	References
**rBCGΔ*sigH***	extracytoplasmic sigma factor SigH	BALB/c and C57BL/6 subcutaneous and retro-orbital	stronger T cell responses more Th1 type cytokine-producing CD4^+^ T cells stronger polyfunctional cytokine-producing CD8^+^ T cells	[132]
**rBCGΔ*zmp1***	zinc metalloprotease	C57BL/6 subcutaneous guinea pigs	stronger mycobacterial antigen presentation by DCs ^a^ superior T cell responses, including T cell proliferation and IFN-γ production increased numbers of mycobacterial antigen-specific IFN-γ^+^CD4^+^ and IFN-γ^+^CD8^+^ T cells. better protection against lung colonization by *M. tuberculosis*	[134,135]
**rBCG*sapM*::Tn**	secreted acid phosphatase	BALB/c subcutaneous F1 intravenous, subcutaneous	significantly lower amounts of *M. tuberculosis* significantly reduced mortality higher levels of IL-1ß, IL-6 and TNF-α by DCs lower numbers of mycobacterial antigen-specific CD4^+^ and CD8^+^ T cells producing IFN-γ in the lymph nodes and spleens enhanced protective effect	[136,137]
**rBCGΔ*2432c***	Aminoglycoside acetyltransferase	C57BL/6 subcutaneous	induced more autophagy and apoptosis of THP-1 macrophages and mycobacterial antigen presentation by DCs significantly stronger protection against *M. tuberculosis* stronger T cell responses, including Th-1 cytokine-producing CD4^+^ and CD8^+^ T cells, CD4^+^ memory T cells, both IFN-γ^+^ effector memory and IL-2^+^ central memory T cells	[138]

^a^ Abbreviation: DC, dendritic cell.

## 7. Conclusions

In an attempt to improve the immune-modulatory effect of BCG, many different molecular engineering approaches have been undertaken, some with very promising immune-enhancing effects, sometimes associated with strongly augmented protective or therapeutic potency against tuberculosis or other diseases. However, almost all studies have been conducted exclusively in mouse models so far. The only exception is VPM1002, the rBCGΔ*ureC::hly* strain that has progressed into clinical development as a novel anti-tuberculosis vaccine candidate. In mice, VPM1002 provided improved protection over pBCG against *M. tuberculosis* challenge, especially against highly virulent Beijing strains [101]. In humans, it was shown to be safe [105], including in HIV-uninfected newborns [106], and able to induce stronger IFN-γ responses and higher frequencies of IFN-γ-producing CD8^+^ T cells than pBCG. Whether this will result in improved protection against tuberculosis in humans remains to be seen in randomized controlled clinical efficacy trials that are currently underway [140]. VPM1002 has also successfully undergone a phase 1 clinical trial for NMIBC treatment in patients with previous failure of the conventional BCG treatment and was found to be well tolerated [108].

All other rBCG have not yet transitioned to human trials. In mice, several rBCG strains have shown improved protection over pBCG against *M. tuberculosis* challenge in addition to their enhanced immunogenicity. Vaccination with rBCG::Ag85B-IL-15 [58] or rBCG::GM-CSF [65] lowered the bacterial burden and pathology in the lungs after *M. tuberculosis* infection compared to rBCG::Ag85B or pBCG vaccination, respectively, illustrating the protective effect of these cytokines, while rBCG producing other cytokines were not more protective than pBCG. Similarly, rBCG::LTA vaccination improved protection against *M. tuberculosis* compared to pBCG and protected mice against highly virulent *M. tuberculosis* Beijing, while pBCG did not [86]. Interestingly, over-expression of some intrinsic BCG antigens, such as Ag85B also resulted in improved protection against *M. tuberculosis* challenge [117]. Other genetic modifications, such as those leading to the expression of SOCS1 antagonists [123] or enhanced c-di-AMP production [130], have also led to significantly improved protection in mice, as have been several deletions of immune-suppressive genes in the BCG chromosome, such as *sigH* [133], *zmp1* [135], *sapM* [136] and *eis* [137]. Since these genetic modifications resulted in improved protection by a variety of different mechanisms, it may be interesting to know whether the combination of all of them within a single rBCG strain would result in a super-efficacious anti-tuberculosis vaccine.

Next to the prevention of tuberculosis, the second most common indication for BCG is intravesical therapy in recurrent NMIBC patients. However, a significant number of patients do not respond to BCG therapy [141]. Therefore, treatment of these patients with an improved rBCG strain may potentially circumvent treatment failure. Intravesical instillation of rBCG secreting certain cytokines, such as IFN-γ [35], IFN-α [43] or IL-15 [57], or various chimeric proteins in which GM-CSF is fused to cancer antigens [67,68,69], has been shown to result in prolonged survival and enhanced tumor growth suppression compared to the installation of pBCG in various mouse models of bladder cancer. Expression of bacterial toxin fragments, such as PTA [93] or increased production of c-di-AMP [131], has also improved the performance of BCG against tumor growth in murine and models.

Finally, rBCG::IL-18 [49] and rBCG::PTA [96] have been demonstrated to reduce Th2 cytokine production and eosophil influx in the lungs, as well as to improve lung function in ovalbumin mouse models of allergic asthma.

Several rBCG developed over the last 30 years have thus shown very promising improved protective or therapeutic potential in rodent models. It would now be important to determine whether these enhanced effects may also translate into improved protective or therapeutic outcomes in human cancer, allergies and/or tuberculosis. Such translational investigations would certainly strongly boost renewed interest in the vaccine for another 100 years or more.

## Data Availability

Not applicable.

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
