# Peer review of "Recombinant BCG to Enhance Its Immunomodulatory Activities"

_vaccines, 2022, doi:10.3390/vaccines10050827_

Round 1

Reviewer 1 Report

The review of BCG and recombinant BCG strains by the authors is exhaustive and thorough.  The material discussed for the myriad of recombinant BCG variants has been succinctly discussed with salient features discussed.  The authors used a similar approach throughout the manuscript. The discussions are brief but give a sense of the experimental outcomes and pathogenic mechanisms.

A table that gave an overview of this detailed review would aid readers.  In that table the key comparison could be: recombinant BCG(appropriate name-cytokine/bacterial toxin/etc) vs BCG--Effect on M. tuberculosis replication, bladder cancer growth/survival, etc (in animal species, cell culture, etc) with emphasis on clear differences between the rBCG and BCG. For example, many of the studies showed clear differences in particular cytokine production but effect on the grow of M. tuberculosis was not tested.

The table could provide the reader with a concise comparison that has been described in the review.

Author Response

Reply to Reviewer 1

Referee #1

The review of BCG and recombinant BCG strains by the authors is exhaustive and thorough.  The material discussed for the myriad of recombinant BCG variants has been succinctly discussed with salient features discussed.  The authors used a similar approach throughout the manuscript. The discussions are brief but give a sense of the experimental outcomes and pathogenic mechanisms.

A table that gave an overview of this detailed review would aid readers.  In that table the key comparison could be: recombinant BCG(appropriate name-cytokine/bacterial toxin/etc) vs BCG--Effect on M. tuberculosis replication, bladder cancer growth/survival, etc (in animal species, cell culture, etc) with emphasis on clear differences between the rBCG and BCG. For example, many of the studies showed clear differences in particular cytokine production but effect on the grow of M. tuberculosis was not tested.

The table could provide the reader with a concise comparison that has been described in the review.

Reply: We thank the reviewer for his/her nice words and agree that a table would be useful here. Accordingly, we prepared 3 Tables with the concise description of the key features of the various recombinant BCG strains.

Table 1 describes recombinant BCG producing cytokines in tuberculosis and non-tuberculosis diseases

Table 2 describes recombinant BCG strains producing bacterial effectors and other non-bacterial immunomodulatory effectors

Table 3 describes rBCG with chromosomal deletions.  

Reviewer 2 Report

The review discusses in a very comprehensive manner recombinant BCG vaccines that have been developed  so far. Different strategies and approaches were described, albeit most of them did not succeeded to  enter into human clinical studies. . Furthermore, it also describes other possible uses of recombinant BCG vaccine, such as cancer, asthma. However, the manuscript has not been prepared carefully -  there are lots of missing symbols (IFN-gamma,  TNF-alpha, etc), which makes it difficult to read. Additionally, some phrases should be checked for the missing general sense, such as, for example: In 939 humans, it was shown to be safe [103], including in HIV-uninfected newborns [104], and 940 able to induce stronger IFN- responses and higher frequencies of IFN--producing 941 CD8+ T cells than pBCG. (line 940). The review would also gained a lot, if the authors prepared a simple diagram illustrating  approaches to develop rBCG  and their final effect (works or not).

Author Response

Reply to Reviewer 2

Referee #2

The review discusses in a very comprehensive manner recombinant BCG vaccines that have been developed  so far. Different strategies and approaches were described, albeit most of them did not succeeded to  enter into human clinical studies. . Furthermore, it also describes other possible uses of recombinant BCG vaccine, such as cancer, asthma. However, the manuscript has not been prepared carefully -  there are lots of missing symbols (IFN-gamma,  TNF-alpha, etc), which makes it difficult to read. Additionally, some phrases should be checked for the missing general sense, such as, for example: In 939 humans, it was shown to be safe [103], including in HIV-uninfected newborns [104], and 940 able to induce stronger IFN- responses and higher frequencies of IFN--producing 941 CD8+ T cells than pBCG. (line 940). The review would also gained a lot, if the authors prepared a simple diagram illustrating  approaches to develop rBCG  and their final effect (works or not).

Reply: We thank the reviewer for his/her comments. The missing symbols and strange sentences were due to the transition of the original manuscript to the template. We have carefully checked the paper for such errors and have corrected them throughout the paper. Instead of a diagram, we have prepared three tables  with the various rBCG approaches and their effects, as also requested by reviewer #1.

Reviewer 3 Report

  1. The objective of the review is diffused. The two points that surfaced from the review are overlapping : i) rBCG as a better BCG vaccine against tuberculosis and ii) rBCG against non-specific pathogens.

 It is hard to follow the objectives of the review. It should be more specific and clear.

2. If the author chooses to elaborate the studies on rBCG for designing a more effective vaccine against tuberculosis they should also discuss the reasons for failure of pBCG. Sensitization to environmental Mycobacterium is one of the main reasons for failure of BCG in some parts of the world. The authors should address how rBCG can overcome this problem.

3) It is also shown that different strains of BCG have differential efficacy of protection (Mol Ther. 2016 Feb; 24(2): 201–203.). Which strain of BCG is best to use for designing rBCG ?

4. While discussing rBCG efficacy against non-specific pathogens, the author should also write about the BCG induced trained immunity.

5. IL-2, IFN, GM-CSF are generalized immunomodulatory molecules for enhancing the immune function. In designing any vaccine (if having insufficient immune response) these are used to enhance immunogenicity.  Elaboration of IL-2, IFN-g, GM-CSF and other functions are unnecessary as these are well known from text book knowledge. Focus of the review should be on rBCG candidates that have been tested against tuberculosis or other infections in animal models or clinical trials. More emphasis should be given on different rBCG vaccines which should also include:

1)AEREAS-401 and AERAS-422, rBCG expressing the cholesterol-binding cytoslysin perfringolysin O (Pfo) should be discussed. (Vaccine (2009) 27(33):4412–23, Eur J Biochem (2002) 269(24):6195–203)

2)rBCG expressing Mtb antigens is also a lead to BCG vaccine research. Needs discussion in the MS.

6.  Is there any specific reason for not including “rBCG strains producing antigens from heterologous pathogens to engineer specific mul- tivalent vaccines”? There are lot of animal studies using HIV, Schistoma, Leishmania, RSV antigen to develop rBCG. I think this discussion is more important as it offers a new platform of vaccine design combining both the specific (adaptive) and non-specific (innate) arm of immunity to offer broad spectrum of protection. Infact rBCG-RSV vaccine has also undergone clinical trials but nothing has been discussed in this review.

7.  rBCG has been used in various studies to combat or provide prime boost for Influenza, RSV, SARS CoV2 infections. However authors have not discussed its utility or failure in these settings.

8.  It would be helpful if various rBCG platforms, their efficacy, studies on mouse/human, references can be provided in a table. Also how rBCG may work better that BCG can be shown as diagrammatic representation.

In current form, manuscript is monotonous and feels like a report.

vii) Authors may consider discussing/including the following research papers

  1. Kaufmann et. Al., 2022, W. Gong et al 2018, A Mansoor et.al., 2021, FE Diaz  2021, AC de Costa 2014, James A Tricas 2010 , TL Oliveira 2017,  AI Kanno 2022, E Mouhoub 2021 and NE Nieuwahuizen 2021  etc etc

Author Response

Reply to Reviewer 3

Referee #3

  1. The objective of the review is diffused. The two points that surfaced from the review are overlapping : i) rBCG as a better BCG vaccine against tuberculosis and ii) rBCG against non-specific pathogens.

 It is hard to follow the objectives of the review. It should be more specific and clear.

Reply: We thank the reviewer for his/her comments. The objective of this review was to summarize the development of recombinant BCG strains with enhanced immune-modulatory effects to be used either as improved anti-tuberculosis vaccines or in immunotherapy for non-mycobacterial diseases. This is reflected in the title and at the end of the abstract. We have now added an additional sentence at the end of the introduction to clarify this objective.

  1. If the author chooses to elaborate the studies on rBCG for designing a more effective vaccine against tuberculosis they should also discuss the reasons for failure of pBCG. Sensitization to environmental Mycobacterium is one of the main reasons for failure of BCG in some parts of the world. The authors should address how rBCG can overcome this problem.

Reply: We agree with the reviewer that sensitization to environmental mycobacteria may be one of the reasons for BCG failure against tuberculosis. However, this is still a matter of debate, and there is no clear evidence for that in humans. Since for the moment the rBCG strains have been tested in tuberculosis models in which the animals were not sensitized to environmental mycobacteria, it is difficult to discuss in this paper the potential effect of exposure to environmental mycobacteria.

3) It is also shown that different strains of BCG have differential efficacy of protection (Mol Ther. 2016 Feb; 24(2): 201–203.). Which strain of BCG is best to use for designing rBCG ?

Reply: Yes, this is true, different BCG strains may have differential efficacies, but this has not been firmly demonstrated in humans. Furthermore, to our knowledge, there are no experimental studies comparing recombinant BCG strains with different genetic backgrounds. We can therefore not elaborate on this issue in this paper.

  1. While discussing rBCG efficacy against non-specific pathogens, the author should also write about the BCG induced trained immunity.

Reply: We have mentioned trained innate immunity in the introduction (line 51-52) and provided a reference (ref. #14). We have also added a new paragraph addressing specifically this issue with a recombinant BCG strain, based on the very recently published article by Kanno et al. (Ref#94see lines 404-411).

  1. IL-2, IFN, GM-CSF are generalized immunomodulatory molecules for enhancing the immune function. In designing any vaccine (if having insufficient immune response) these are used to enhance immunogenicity.  Elaboration of IL-2, IFN-g, GM-CSF and other functions are unnecessary as these are well known from text book knowledge. Focus of the review should be on rBCG candidates that have been tested against tuberculosis or other infections in animal models or clinical trials. More emphasis should be given on different rBCG vaccines which should also include:

1)AEREAS-401 and AERAS-422, rBCG expressing the cholesterol-binding cytoslysin perfringolysin O (Pfo) should be discussed. (Vaccine (2009) 27(33):4412–23, Eur J Biochem (2002) 269(24):6195–203)

2)rBCG expressing Mtb antigens is also a lead to BCG vaccine research. Needs discussion in the MS.

Reply: we have already discussed the AERAS-401 and AERAS-422 vaccine candidates and cited the relevant papers (see Refs. #110 and #114). There is an entire chapter on the recombinant BCG strains that produce perfringolysin O.

  1. Is there any specific reason for not including “rBCG strains producing antigens from heterologous pathogens to engineer specific mul- tivalent vaccines”? There are lot of animal studies using HIV, Schistoma, Leishmania, RSV antigen to develop rBCG. I think this discussion is more important as it offers a new platform of vaccine design combining both the specific (adaptive) and non-specific (innate) arm of immunity to offer broad spectrum of protection. Infact rBCG-RSV vaccine has also undergone clinical trials but nothing has been discussed in this review.

Reply: As indicated in the title, the abstract and introduction, we have focused this paper on molecules that potentially may modify or enhance the immune-modulatory properties of BCG, as this was the objective of the review. Discussing recombinant BCG strains that produce these heterologous antigens would be an entirely different focus and have recently been discussed in the excellent paper by Mouhoub et al. (by the way, kindly provided by this reviewer), which we have now cited in the revised version of the manuscript. We added a sentence at the end of the introduction to clarify this point (Ref#17 see lines 59-60).

  1. rBCG has been used in various studies to combat or provide prime boost for Influenza, RSV, SARS CoV2 infections. However authors have not discussed its utility or failure in these settings.

Reply: We are not aware of studies where recombinant BCG strains have been developed to prime or boost innate immunity against influenza, RSV or SARS-CoV2 infections. Such studies have been done (or are underway) with non-recombinant BCG, which is not the purpose of this paper.

  1. It would be helpful if various rBCG platforms, their efficacy, studies on mouse/human, references can be provided in a table. Also how rBCG may work better that BCG can be shown as diagrammatic representation.

Reply: in agreement with the reviewer, we have now prepared Tables 1-3 summarizing these elements, as also requested by the other reviewers.

In current form, manuscript is monotonous and feels like a report.

vii) Authors may consider discussing/including the following research papers

  1. Kaufmann et. Al., 2022, W. Gong et al 2018, A Mansoor et.al., 2021, FE Diaz  2021, AC de Costa 2014, James A Tricas 2010 , TL Oliveira 2017,  AI Kanno 2022, E Mouhoub 2021 and NE Nieuwahuizen 2021  etc etc

Reply : As stated above, our article focuses on recombinant BCG strains for enhanced immune responses or immune-modulation by BCG, and we are not sure what some of the proposed refences might add. The paper by Kaufmann et al in Cell Rep., 2022 shows protection against IAV with non-recombinant BCG. The paper by Diaz et al Front. Immunol. 2021 is about a recombinant strain that produces the RSV N protein to protect calves against RSV. The paper by Gong et al. in Hum Vaccin Immunother, 2018 is a review paper on novel tuberculosis vaccines and not a research paper. The paper by Triccas in Bioeng Bugs, 2010 is also a review paper, not a research paper, as is the paper by Oliveira in Appl Microbiol Biotechnol, 2017 and the paper by Mouhoub et al in Front. Immunol, 2021.

However, we have cited the latter in the revised version of the manuscript, and thank the reviewer for turning our attention to this paper.

We also have now included the paper by Kanno et al., which came out while this article was completed. Again, we thank the reviewer for this paper. It prompted us to add a paragraph on trained innate immunity by recombinant BCG (Ref#94 see lines 404-411)

We could not find the papers by Mansoor et al., 2021 and by Nieuwahuizen, 2021.

Round 2

Reviewer 3 Report

Thanks for addressing the concerns.